# Seeking Flat Minima with Mean Teacher on Semi- and Weakly-Supervised Domain Generalization for Object Detection

## Abstract

Object detectors do not work well when domains largely differ between training and testing data. To overcome this domain gap in object detection without requiring expensive annotations, we consider two problem settings: semi-supervised domain generalizable object detection (SS-DGOD) and weakly-supervised DGOD (WS-DGOD). In contrast to the conventional domain generalization for object detection that requires labeled data from multiple domains, SS-DGOD and WS-DGOD require labeled data only from one domain and unlabeled or weakly-labeled data from multiple domains for training. In this paper, we show that object detectors can be effectively trained on the two settings with the same Mean Teacher learning framework, where a student network is trained with pseudo-labels output from a teacher on the unlabeled or weakly-labeled data. We provide novel interpretations of why the Mean Teacher learning framework works well on the two settings in terms of the relationships between the generalization gap and flat minima in parameter space. On the basis of the interpretations, we also show that incorporating a simple regularization method into the Mean Teacher learning framework leads to flatter minima. The experimental results demonstrate that the regularization leads to flatter minima and boosts the performance of the detectors trained with the Mean Teacher learning framework on the two settings.

## 1 Introduction

Object detection has been attracting much attention because it has practically useful applications such as in autonomous driving. Object detectors have performed tremendously well on commonly used benchmark datasets for object detection, such as MSCOCO (Lin et al., 2014) and PASCAL VOC (Everingham et al., 2010). However, such performance significantly drops when they are deployed on unseen domains, i.e., when the training and testing domains are different. For example, Inoue et al. (Inoue et al., 2018) reported a performance drop caused by the difference in image styles, and Li et al. (Li et al., 2022) showed one caused by the weather and time difference in the images captured with car-mounted cameras.

To solve this problem, many researchers have been exploring unsupervised domain adaptive object detection (UDA-OD) (Deng et al., 2021; Li et al., 2022; Chen et al., 2022). On UDA-OD, we train object detectors using source domain data with ground-truth labels (bounding boxes and class labels) and unlabeled target domain data to adapt the detectors to the target domain. However, in the real world, target domain data cannot always be accessed in the training phase.

Domain generalizable object detection (DGOD) is another common problem setting for solving the problem of the performance drop caused by the domain gaps (Lin et al., 2021; Zhang et al., 2022a). On DGOD, we train object detectors using labeled data from multiple domains so that the detectors work well on unseen domains. However, it is labor-intensive to collect these data for object detection because both bounding boxes and class labels for all objects in the images must be annotated. Although single-DGOD (Wu & Deng, 2022; Fan et al., 2023; Vidit et al., 2023; Wang et al., 2021b; 2023a; Lee et al., 2024), on which we train object detectors to generalize unseen domains using labeled data from one single domain, has been investigated, the performance gain is still limited.

In this paper, we tackle two tasks as more realistic settings: i) semi-supervised DGOD (SS-DGOD) (Malakouti & Kovashka, 2023) and ii) weakly-supervised DGOD (WS-DGOD). The goal of SS-DGOD is to generalize object detectors to unseen domains using labeled data only from one single domain and unlabeled data from multiple domains. Note that the target domain data are not included in the training data. On WS-DGOD, we use weakly labeled data from multiple domains instead of the unlabeled data in SS-DGOD. "Weakly labeled" means that we have only image-level labels that show the existence of each class in each training image and do not have bounding box annotations. To the best of our knowledge, this is the first attempt to tackle WS-DGOD. We show that object detectors can be effectively trained on the two settings with the same Mean Teacher learning framework, where a student network is trained with pseudo-labels output from a teacher on the unlabeled or weakly labeled data, and the teacher network is updated as the exponential moving average (EMA) of the student.

Not only do we experimentally demonstrate the good performance of the Mean Teacher learning framework, but also provide novel interpretations of why the Mean Teacher learning framework works well on these two settings in terms of the relationship between generalization ability and flat minima in parameter space. These interpretations are based on our findings that the two key components of the Mean Teacher learning framework, i) EMA update and ii) learning from pseudo-labels, lead to flat minima during the training. In the research area of domain generalization, it has been shown both theoretically and empirically that neural networks with flatter minima in parameter space have better generalization ability to unseen domains (Foret et al., 2021; Chaudhari et al., 2017; Cha et al., 2021; Izmailov et al., 2018; Caldarola et al., 2022; Wang et al., 2023b; Kaddour et al., 2022; Zhang et al., 2023).

On the basis of the interpretations, we also show that incorporating a simple regularization method into the Mean Teacher learning framework leads to flatter minima. Specifically, because the teacher and the student have similar loss values around the flat minima, we introduce an additional loss term so that the output from the student network becomes similar to that from the teacher network. The experimental results demonstrate that the detectors trained with the Mean Teacher learning framework perform well for unseen test domains on the two settings. We show that the simple yet effective regularization leads to flatter minima and boosts the performance of those detectors.

It is noteworthy that our aim is not to propose an entirely new method or surpass the state-of-the-art methods. Instead, our contributions are summarized as follows:

- We show that object detectors can be effectively trained on the SS-DGOD and WS-DGOD settings with the same Mean Teacher learning framework.

- We provide interpretations of why the detectors trained with the Mean teacher learning framework achieve robustness to unseen test domains in terms of the flatness of minima in parameter space, based on our novel finding that the Mean Teacher leads to flat minima.

- On the basis of the interpretations, we introduce a simple regularization method into the Mean Teacher learning framework to achieve flatter minima.

- We are the first to tackle the WS-DGOD setting.

## 2 Problem Settings

We formally describe the two problem settings of SS-DGOD and WS-DGOD. Their goal is to obtain object detectors that perform well on unseen target domain data $\mathcal{D}_t = \{X_t\}$, where $X_t$ is a set of images from the target domain.

On SS-DGOD, we have labeled data from a source domain $\mathcal{D}_{s_1} = \{(X_{s_1}, B_{s_1}, C_{s_1})\}$ and unlabeled data from multiple source domains $\mathcal{D}_{s_i} = \{X_{s_i}\}_{i=2}^{N_D}$ in the training phase. Here, $X_{s_1} = \{x_{s_1}^j\}_{j=1}^{N_{s_1}}$ is a set of $N_{s_1}$ images from domain $s_1$. $B_{s_1} = \{b_{s_1}^j\}_{j=1}^{N_{s_1}}$ and $C_{s_1} = \{c_{s_1}^j\}_{j=1}^{N_{s_1}}$ are the corresponding bounding boxes and object-class labels, respectively. $s_i$ is the $i$-th source domain, and $N_\mathcal{D}$ is the number of the source domains. We assume that the data distributions differ between the domains, i.e., $P(X_{s_1}) \neq P(X_{s_2}) \neq \cdots P(X_{s_{N_D}}) \neq P(X_t)$.

Figure 1: Comparisons of the problem settings.

Table 1: Formal comparisons of SS-DGOD, WS-DGOD, and related problem settings. DGOD stands for domain generalizable object detection, and SS-DGOD and WS-DGOD are semi-supervised DGOD and weakly-supervised DGOD, respectively. UDA-OD is unsupervised domain adaptive object detection, and WSDA-OD is weakly-supervised domain adaptive object detection.

| task | train data | test data |
|------|------------|-----------|
| Single-DGOD | $\mathcal{D}_{s_1} = \{(X_{s_1}, B_{s_1}, C_{s_1})\}$ | $\mathcal{D}_t = \{X_t\}$ |
| **SS-DGOD** | $\mathcal{D}_{s_1} = \{(X_{s_1}, B_{s_1}, C_{s_1})\}, \mathcal{D}_{s_i} = \{X_{s_i}\}_{i=2}^{N_D}$ | $\mathcal{D}_t = \{X_t\}$ |
| **WS-DGOD** | $\mathcal{D}_{s_1} = \{(X_{s_1}, B_{s_1}, C_{s_1})\}, \mathcal{D}_{s_i} = \{(X_{s_i}, C_{s_i})\}_{i=2}^{N_D}$ | $\mathcal{D}_t = \{X_t\}$ |
| DGOD | $\mathcal{D}_{s_i} = \{(X_{s_i}, B_{s_i}, C_{s_i})\}_{i=1}^{N_D}$ | $\mathcal{D}_t = \{X_t\}$ |
| UDA-OD | $\mathcal{D}_{s_1} = \{(X_{s_1}, B_{s_1}, C_{s_1})\}, \mathcal{D}_t = \{X_t\}$ | $\mathcal{D}_t = \{X_t\}$ |
| WSDA-OD | $\mathcal{D}_{s_1} = \{(X_{s_1}, B_{s_1}, C_{s_1})\}, \mathcal{D}_t = \{(X_t, C_t)\}$ | $\mathcal{D}_t = \{X_t\}$ |

On WS-DGOD, we use labeled data from a source domain $\mathcal{D}_{s_1} = \{(X_{s_1}, B_{s_1}, C_{s_1})\}$ and weakly labeled data from multiple domains $\mathcal{D}_{s_i} = \{(X_{s_i}, C_{s_i})\}_{i=2}^{N_D}$ for training.

Table 1 compares SS-DGOD and WS-DGOD with related problem settings (Single-DGOD, DGOD, and UDA-OD). As discussed in Sec. 1, DGOD requires labeled data from multiple domains $\mathcal{D}_{s_i} = \{(X_{s_i}, B_{s_i}, C_{s_i})\}_{i=1}^{N_D}$, but those data are sometimes hard to prepare due to the high annotation cost. In contrast, SS-DGOD (or WS-DGOD) requires labeled data from one domain $\mathcal{D}_{s_1} = \{(X_{s_1}, B_{s_1}, C_{s_1})\}$ and unlabeled data $\mathcal{D}_{s_i} = \{X_{s_i}\}_{i=2}^{N_D}$ (or weakly labeled data $\mathcal{D}_{s_i} = \{(X_{s_i}, C_{s_i})\}_{i=2}^{N_D}$), which are easier to obtain. Therefore, SS-DGOD and WS-DGOD are more practical settings than DGOD. By using those data, we aim to better generalize object detectors to the unseen target domain data $\mathcal{D}_t = \{X_t\}$ than on Single-DGOD. Although another type of SS-DGOD, where a portion of the samples are labeled in each source domain, can also be defined, we will leave it as part of our future work (see Appendix D.1).

Another related setting is weakly-supervised domain adaptive object detection (WSDA-OD), a.k.a., cross-domain weakly-supervised object detection (Inoue et al., 2018; Hou et al., 2021; Xu et al., 2022; Tang et al., 2023), which requires weakly-labeled target domain data $\mathcal{D}_t = \{(X_t, C_t)\}$ for training. Unlike on UDA-OD and WSDA-OD, we can train the detectors even when the unlabeled or weakly-labeled target domain data ($\mathcal{D}_t = \{X_t\}$ or $\mathcal{D}_t = \{(X_t, C_t)\}$) are not accessible.

# 3 Related Work

## 3.1 Domain Generalization for Image Classification

Many methods have been proposed for domain generalization on image classification tasks as summarized in recent survey papers (Zhou et al., 2022; Wang et al., 2022a). Among a variety of domain generalization methods, finding flat minima is one of the most common approaches (Foret et al., 2021; Chaudhari et al., 2017; Cha et al., 2021; Izmailov et al., 2018; Caldarola et al., 2022; Wang et al., 2023b; Kaddour et al., 2022; Zhang et al., 2023; 2024). Those studies empirically and theoretically showed that finding flat minima in parameter space results in a better generalization ability. Izmailov et al. (2018) and Cha et al. (2021) demonstrated that empirical risk minimization (ERM) with stochastic gradient descent (SGD) converges to the vicinity of a flat minimum, and averaging the parameter weights over a certain number of training steps/epochs results in reaching the flat minimum. Inspired by these findings, we reveal that the Mean Teacher learning framework leads to flat minima, and thus can obtain good generalization ability.

## 3.2 Domain Generalization for Object Detection

Domain generalization for object detection has not been widely explored, compared with image classification. Lin et al. (2021) proposed a method for disentangling domain-specific and domain-invariant features by adversarial learning on both image-level and instance-level features for DGOD. Liu et al. (2020) investigated DGOD in underwater object detection and proposed DG-YOLO. For Single-DGOD, Wang et al. (2021b) proposed a self-training method that uses the temporal consistency of objects in videos. Wu & Deng (2022) proposed a method for disentangling domain-invariant features by contrastive learning and self-distillation. Fan et al. (2023) proposed perturbing the channel statistics of feature maps, which can be interpreted as data augmentation of image styles to a variety of domains. Wang et al. (2023a) proposed a disentangle method on frequency space for object detection from unmanned aerial vehicles. Vidit et al. (2023) proposed an augmentation method using a pre-trained vision-language model (CLIP) with textual prompts.

Unlike the above methods, as discussed in Sects. 1 and 2, we tackle SS-DGOD (Semi-Supervised Domain Generalization for Object Detection) and a new problem setting called WS-DGOD (Weakly-Supervised Domain Generalization for Object Detection). The most closely related to our work is Malakouti & Kovashka (2023)'s work. They tackled SS-DGOD and proposed a language-guided alignment method. However, the limitation of their method is that it requires a backbone network that was pre-trained on vision-and-language tasks. Our experiments show that the object detectors trained with the Mean Teacher learning framework and our regularization outperform their method when the same backbone is used.

## 3.3 Semi-supervised Domain Generalization

There are a few methods that use both labeled and unlabeled data for domain generalization (SSDG) on image classification (Zhang et al., 2022b; Zhou et al., 2023b; Lin et al., 2024). Zhang et al. (2022b) proposed an unsupervised pre-training method called DARLING, which performs contrastive learning on unlabeled images to obtain domain-irrelevant feature representation. Zhou et al. (2023b) extended a semi-supervised learning method called FixMatch (Sohn et al., 2020) to SSDG.

In contrast to those studies, we tackle SSDG for object detection. We also tackle the "weakly-labeled" setting (i.e., WS-DGOD), which has not been explored even for image classification.

## 3.4 Mean Teacher Learning Framework

Mean Teacher learning framework was originally proposed for semi-supervised image classification (Tarvainen & Valpola, 2017). Several studies have investigated the use of the Mean Teacher learning framework for a variety of tasks such as domain generalization on image classification (Yang et al., 2021), (in-domain) weakly-supervised object detection (Wang et al., 2022b), (in-domain) semi-supervised object detection (Mi et al., 2022), UDA-OD (Deng et al., 2021; Li et al., 2022; He et al., 2022; Deng et al., 2023; Kennerley et al., 2024), and UDA for semantic segmentaion (Araslanov & Roth, 2021; Wang et al., 2021a; Hoyer et al., 2022;

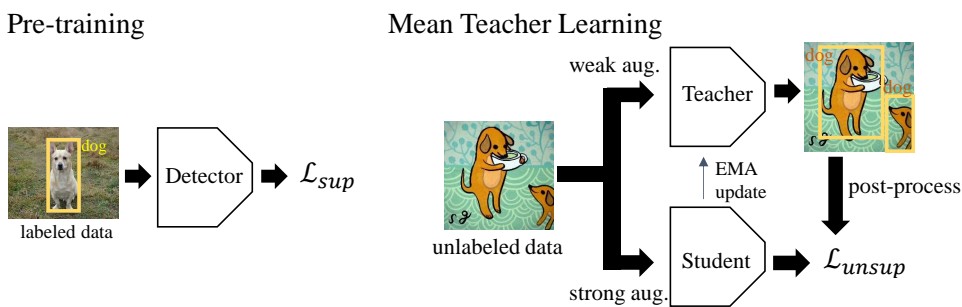

Figure 2: Training framework.

Zhang et al., 2021). Lee et al. (2023) provided a theoretical analysis of the Mean Teacher learning framework on masked image modeling pretext tasks for semi-supervised image classification. We show that the Mean Teacher learning framework also works well on different settings (SS-DGOD and WS-DGOD), provide their interpretations, and introduce a simple regularization method to lead to flatter minima.

## 4 Training Method

### 4.1 Overview and Key Idea

On both SS-DGOD and WS-DGOD, our goal is to obtain object detectors that work well on the unseen target domain data $\mathcal{D}_t = \{X_t\}$. Gulrajani & Lopez-Paz (2021) reported that if carefully implemented, empirical risk minimization (i.e., the image classifier simply trained with supervised learning on multiple domains) outperformed state-of-the-art domain generalization methods on several benchmark datasets for image classification. Following this important finding, we expect similar behavior on object detection and aim to train an object detector on multiple domains $\mathcal{D}_{s_i}(i = 1, \cdots, N_D)$. However, we have no ground-truth labels (or have only weak labels) for $\mathcal{D}_{s_i}(i = 2, \cdots, N_D)$ although ground-truth labels are available for $\mathcal{D}_{s_1}$. Therefore, the question is how to train a detector on those domains. Our solution is to use the Mean Teacher learning framework for object detection (Li et al., 2022; Chen et al., 2022) shown in Fig. 2, where we have two networks (teacher and student) with the same structure and train the student network using the pseudo-labels generated by the teacher network. Note that this Mean Teacher learning framework can be applied to any object detector, but we hereafter describe the loss functions of FasterRCNN (Ren et al., 2015) as an example for ease of explanation.

### 4.2 Pre-training

If we start the Mean Teacher learning from randomly initialized parameters, the teacher network cannot output reliable pseudo labels. Therefore, we first perform supervised learning with the labeled data of one source domain $\mathcal{D}_{s_1} = \{(X_{s_1}, B_{s_1}, C_{s_1})\}$.

$$
\begin{aligned}
\mathcal{L}_{s_1}^{\text{sup}}(\theta) = {} & \mathcal{L}_{\text{RPN}}^{\text{cls}}(\theta, X_{s_1}, B_{s_1}, C_{s_1}) + \mathcal{L}_{\text{RPN}}^{\text{reg}}(\theta, X_{s_1}, B_{s_1}, C_{s_1}) \\
& + \mathcal{L}_{\text{RoI}}^{\text{cls}}(\theta, X_{s_1}, B_{s_1}, C_{s_1}) + \mathcal{L}_{\text{RoI}}^{\text{reg}}(\theta, X_{s_1}, B_{s_1}, C_{s_1}),
\end{aligned}
\tag{1}
$$

where $\mathcal{L}_{\text{RPN}}^{\text{cls}}$ and $\mathcal{L}_{\text{RPN}}^{\text{reg}}$ are the classification and regression losses for region proposal networks (RPN), respectively. $\mathcal{L}_{\text{RoI}}^{\text{cls}}$ and $\mathcal{L}_{\text{RoI}}^{\text{reg}}$ are those for RoIhead. We initialize both the teacher and student networks with the parameters $\theta^* = \arg\min_\theta \mathcal{L}_{s_1}^{\text{sup}}(\theta)$ obtained from this pre-training.

### 4.3 Mean Teacher Learning

#### 4.3.1 Generate Pseudo-labels

Because we have no ground-truth labels (or have only weak labels) for the other source domains $\mathcal{D}_{s_i}(i = 2, \cdots, N_D)$, we generate pseudo labels using the teacher network. Specifically, we perform weak data augmentation to the unlabeled (or weakly-labeled) image $x_{s_i}^j$ and input it into the teacher network. We denote the output from the teacher as $\{(\hat{b}_{s_i}^{jr}, \hat{p}_{s_i}^{jr})\}_{r=1}^{N_R}$, where $\hat{b}_{s_i}^{jr}$ and $\hat{p}_{s_i}^{jr}$ are the predicted bounding box and class probabilities for the $r$-th region of interests (RoI) in the $j$-th image, respectively, and $N_R$ is the number of output RoIs.

In the case of SS-DGOD, we simply perform post-processing $f_{post}$ to $(\hat{b}_{s_i}^{jr}, \hat{p}_{s_i}^{jr})$ and obtain the pseudo label $(\bar{b}_{s_i}^{jr}, \bar{c}_{s_i}^{jr}) = f_{\text{post}}(\hat{b}_{s_i}^{jr}, \hat{p}_{s_i}^{jr})$. Post-processing $f_{post}$ indicates a simple thresholding function if we use "hard" pseudo labels like (Li et al., 2022) and indicates a sharpening function if we use "soft" pseudo labels like (Chen et al., 2022).

In the case of WS-DGOD, we perform the refinement process of applying the weak labels to the predicted class probabilities $\hat{p}_{s_i}^{jr}$ immediately before post-processing $f_{post}$ to obtain more accurate pseudo labels as follows:

$$(\bar{b}_{s_i}^{jr}, \bar{c}_{s_i}^{jr}) = f_{\text{post}}(\hat{b}_{s_i}^{jr}, \hat{p}_{s_i}^{jr}), \quad \hat{p}_{s_i}^{jr}(k) = \begin{cases} \hat{p}_{s_i}^{jr}(k) & \text{if } k \in c_{s_i}^j \\ 0 & \text{otherwise} \end{cases} \tag{2}$$

where $\hat{p}_{s_i}^{jr}(k)$ is the predicted class probability for the $k$-th class. Using the weak label $c_{s_i}^j$, Eq. (2) makes the predicted probability zero at each RoI if the $k$-th class does not exist in the $j$-th image.

#### 4.3.2 Update Student

Now we have the pseudo labels $\bar{B}_{s_i} = \{\bar{b}_{s_i}^j\}_{j=1}^{N_{s_i}}$ and $\bar{C}_{s_i} = \{\bar{c}_{s_i}^j\}_{j=1}^{N_{s_i}}$ and train the student network with them.

We perform strong data augmentations to the image $x_{s_i}^j$ and input it into the student network. In domain $s_1$, because the ground-truth labels are available, we update the student by backpropagating loss $\mathcal{L}_{s_1}^{\text{sup}}$ in Eq. (1). In the other domains $s_i(i = 2, \cdots, N_D)$, we calculate loss $\mathcal{L}_{s_i}^{\text{unsup}}$ using the pseudo labels and backpropagate it to update the student. In summary, we update the parameters of student $\theta^{\text{student}}$ with loss $\mathcal{L}^{\text{student}}$ as follows:

$$\theta^{\text{student}} \leftarrow \theta^{\text{student}} - \nabla_\theta \mathcal{L}^{\text{student}}(\theta), \quad \mathcal{L}^{\text{student}}(\theta) = \mathcal{L}_{s_1}^{\text{sup}}(\theta) + \sum_{i=2}^{N_D} \mathcal{L}_{s_i}^{\text{unsup}}(\theta) \tag{3}$$

$$\begin{aligned} \mathcal{L}_{s_i}^{\text{unsup}}(\theta) = &\mathcal{L}_{\text{RPN}}^{\text{cls}}(\theta, X_{s_i}, \bar{B}_{s_i}, \bar{C}_{s_i}) + \mathcal{L}_{\text{RPN}}^{\text{reg}}(\theta, X_{s_i}, \bar{B}_{s_i}, \bar{C}_{s_i}) \\ &+ \mathcal{L}_{\text{RoI}}^{\text{cls}}(\theta, X_{s_i}, \bar{B}_{s_i}, \bar{C}_{s_i}) + \mathcal{L}_{\text{RoI}}^{\text{reg}}(\theta, X_{s_i}, \bar{B}_{s_i}, \bar{C}_{s_i}). \end{aligned} \tag{4}$$

#### 4.3.3 Update Teacher

Similar to previous studies (Chen et al., 2022; Li et al., 2022), we do not update the parameters of the teacher $\theta^{\text{teacher}}$ by backpropagation to obtain stable pseudo labels. Instead, we update them by the exponential moving average (EMA) of the parameters of the student network $\theta^{\text{teacher}} \leftarrow \alpha\theta^{\text{teacher}} + (1 - \alpha)\theta^{\text{student}}$. Here, $\alpha$ is a hyperparameter to control the update speed.

## 5 Why Does Mean Teacher Become Robust to Unseen Domains?

We provide novel interpretations of why the Mean Teacher learning framework works well on SS-DGOD and WS-DGOD settings in terms of the relationship between generalization ability and flat minima in parameter space. We show that the two key components of the Mean Teacher learning framework, i) EMA update and ii) learning from pseudo labels, lead to flat minima during the training.

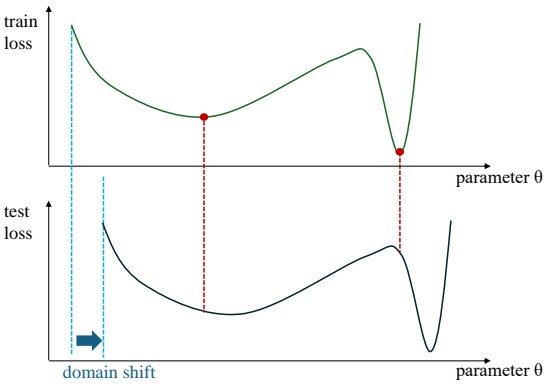

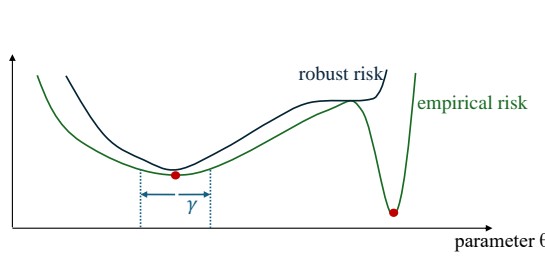

Figure 3: Intuitive interpretation of flat minimum and its robustness to domain shift.

Figure 4: Empirical and robust risks.

## 5.1 Definition

We define an empirical risk as $\mathcal{E}_{\mathrm{ER}}(\theta) := \sum_{i=1}^{N_D} \mathcal{L}_{s_i}^{\sup}(\theta)$ when we assume that ground-truth labels are available on all the training domains. A risk at the target domain is defined as $\mathcal{E}_t(\theta) := \mathcal{L}_t^{sup}(\theta)$. The goal is to minimize the test risk $\mathcal{E}_t(\theta)$ by only solving the empirical risk minimization (ERM), i.e., $\min_\theta \mathcal{E}_{\mathrm{ER}}(\theta)$. Hereafter, we use the terms *risk* and *loss* interchangeably.

## 5.2 Preliminary Knowledge

Previous studies for domain generalization demonstrated both theoretically and empirically that neural networks with flatter minima in parameter space exhibit superior generalization ability to unseen domains (Foret et al., 2021; Chaudhari et al., 2017; Cha et al., 2021; Izmailov et al., 2018; Caldarola et al., 2022; Wang et al., 2023b; Kaddour et al., 2022; Zhang et al., 2023). Fig. 3 shows its intuitive interpretation. We can see that when there is a domain shift between training and testing, the flat minimum of the training loss results in a lower test loss than the sharp minimum.

Cha et al. (2021) theoretically revealed the relationship between the flat minima and generalization gap (i.e., performance drop by domain shift). We briefly describe the theorem for the subsequent explanation. We consider the worst-case loss within neighbor regions in parameter space, which is defined as a robust risk $\mathcal{E}_{RR}^\gamma(\theta) := \max_{\|\Delta\| \le \gamma} \mathcal{E}_{ER}(\theta + \Delta)$. Here, $\gamma$ is the radius of the neighbor region. As shown in Fig. 4, when $\gamma$ is sufficiently large, sharp minima of the empirical risk are not minima of the robust risk. In contrast, the minima of the robust risk (i.e., $\arg\min_\theta \mathcal{E}_{RR}^\gamma(\theta)$) are also minima in the flat regions of the empirical risk. The following theorem shows the relationship between the optimal solution of robust risk minimization (RRM) and the generalization gap in the test domain:

**Theorem** (from (Cha et al., 2021)). *Consider a set of $N$ covers $\{\Theta_k\}_{k=1}^N$ such that the parameter space $\Theta \subset \cup_k^N \Theta_k$ where $\mathrm{diam}(\Theta) := \sup_{\theta,\theta' \in \Theta} \|\theta - \theta'\|_2$, $N := \lceil (\mathrm{diam}(\Theta)/\gamma)^d \rceil$ and $d$ is dimension of $\Theta$. Let $\theta^\gamma$ denote the optimal solution of the RRM, i.e., $\theta^\gamma := \arg\min_\theta \mathcal{E}_{RR}^\gamma(\theta)$, and let $v_k$ and $v$ be VC dimensions of each $\Theta_k$ and $\Theta$, respectively. Then, the gap between the optimal test loss, $\min_{\theta'} \mathcal{E}_t(\theta')$, and the test loss of $\theta^\gamma$, $\mathcal{E}_t(\theta^\gamma)$, has the following bound with probability of at least $1 - \delta$.*

$$
\begin{aligned}
\mathcal{E}_t(\theta^\gamma) - \min_{\theta'} \mathcal{E}_t(\theta') &\le \mathcal{E}_{RR}^\gamma(\theta^\gamma) - \min_{\theta''} \mathcal{E}_{ER}(\theta'') + \frac{1}{N_D} \sum_{i=1}^{N_D} \mathrm{Div}(s_i, t) \\
&+ \max_{k \in [1,N]} \sqrt{\frac{v_k \ln(m/v_k) + \ln(2N/\delta)}{m}} + \sqrt{\frac{v \ln(m/v) + \ln(2/\delta)}{m}},
\end{aligned}
\tag{5}
$$

*where $m$ is the number of training samples and $\mathrm{Div}(s_i, t) := 2\sup_A |\mathbb{P}_{s_i}(A) - \mathbb{P}_t(A)|$ is a divergence between two distributions.*

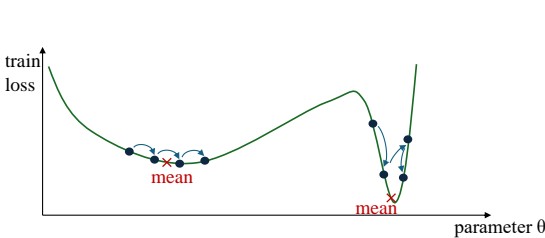

Figure 5: Intuitive interpretation of difference between loss values of trajectory of student and their mean (teacher).

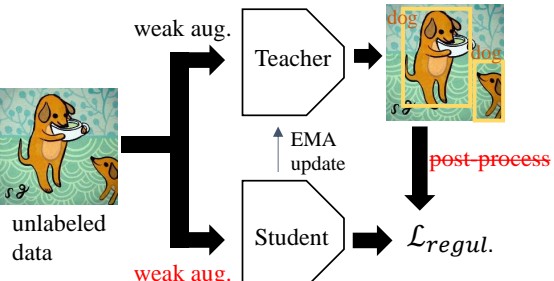

Figure 6: Overview of regualization method.

For its proof, see (Cha et al., 2021). From the theorem, we can interpret that the gap between the RRM and ERM (i.e., $\mathcal{E}^{\gamma}_{RR}(\theta^{\gamma}) - \min_{\theta''} \mathcal{E}_{ER}(\theta''))$ upper bounds the generalization gap in the test domain (i.e., $\mathcal{E}_t(\theta^{\gamma}) - \min_{\theta'} \mathcal{E}_t(\theta'))$. Intuitively, as shown in Fig 4, the gap between the RRM and ERM narrows at flat regions of ERM. Therefore, we can interpret that lowering the gap leads to flat minima of ERM and results in better generalization performance on the target domain.

### 5.3 EMA Update

We explain why the EMA update in the Mean Teacher learning framework leads to flat minima. Stephan et al. (2017) showed that optimizing with constant SGD (i.e., SGD with a fixed learning rate) converges to a Gaussian distribution centered on the optimum. On the basis of this finding, Izmailov et al. (2018) and Cha et al. (2021) showed that the ERM with SGD converges to the marginal of a flat minimum, and averaging the weights of the parameters over some training steps/epochs leads to the flat minima. To avoid overfitting, Izmailov et al. (2018) and Cha et al. (2021) proposed sophisticated algorithms called SWA and SWAD for averaging the weights, and Arpit et al. (2022) introduced a carefully designed averaging strategy called SMA. In contrast to them, we found that a simple EMA also leads to flat minima, even without relying on those averaging algorithms. While the theoretical benefits of weight averaging have been discussed in previous works, this specific finding has not been provided. The experiments presented in Sec. 7 show that the teacher network with only the EMA update of the student (i.e., without pseudo labeling) as shown in Eqs. (6-7) can reach flatter minima and perform better than the student.

$$\theta^{\text{student}} \leftarrow \theta^{\text{student}} - \nabla_\theta \mathcal{L}^{\text{student}}(\theta), \quad \mathcal{L}^{\text{student}}(\theta) = \mathcal{L}^{\text{sup}}_{s_1}(\theta) \tag{6}$$

$$\theta^{\text{teacher}} \leftarrow \alpha\theta^{\text{teacher}} + (1 - \alpha)\theta^{\text{student}}, \tag{7}$$

### 5.4 Learning from Pseudo Labels

We explain why learning from pseudo-labels in the Mean Teacher learning framework leads to flat minima. Assuming that the pseudo-labels from the teacher are accurate enough (i.e., similar enough to ground truth), $\mathcal{L}^{unsup}_{s_i}$ in Eq. (3) can be approximated by $\mathcal{L}^{sup}_{s_i}$, and we can regard the student network as the ERM in Sec. 5.1. On the other hand, as explained in Sec. 5.3 and shown in the experiments, because the teacher network updated with EMA has a better ability to reach flat minima than the student, the teacher can obtain less robust risk than the student, and we can regard the teacher as the robust risk minimizer. Therefore, from Eq. (5), the smaller the difference between the losses of the teacher and student, the smaller the generalization gap in the target domain is. Fig. 5 shows its intuitive interpretations. At the flat region, the trajectory of the student over the training steps and their mean (teacher) have similar loss values. In contrast, there is a large difference between the loss values of the trajectory of the student and their mean at the sharp valley.

Next, we show that learning from pseudo-labels in the Mean Teacher learning framework makes the losses of the student and teacher similar. Because the student is trained with the output from the teacher as

pseudo-ground truth, the training promotes the outputs from the student similar to those from the teacher. When we use monotonically increasing/decreasing functions with respect to the outputs as loss functions $\mathcal{E}$ (e.g., cross-entropy loss $\mathcal{E}(p) = p_{gt} \log(p)$), the more similar the outputs are, the more similar the loss values are, as shown below:

**Proposition.** *Assume $p_1 < p_2 < p_3 \in \mathbb{R}$, and $\mathcal{E}(p) : \mathbb{R} \to \mathbb{R}$ is a monotonically increasing/decreasing function of $p$. Then, $|\mathcal{E}(p_3) - \mathcal{E}(p_2)| < |\mathcal{E}(p_3) - \mathcal{E}(p_1)|$ holds.*

Let us consider $p_3$ as the teacher's output, and $p_2$ and $p_1$ as the outputs of the student. Since $p_2$ is closer to $p_3$ than $p_1$, the loss of $p_2$ becomes more similar to the loss of $p_3$ than that of $p_1$. Therefore, we can interpret that learning from pseudo-labels align the outputs from the student to be similar to those from the teacher, thereby aligning the loss values, consequently leading to flat minima.

## 6 Regularization for Flatter Minima

### 6.1 Method

As discussed in Sec. 5, when the output from the student and teacher are similar, the networks tend to reach flat minima. To this end, we introduce a simple regularization method to make the two networks' outputs more similar by training the student using raw outputs from the teacher.

Fig. 6 shows an overview of the method. The concept is to apply regularization so that the outputs from the two networks are similar for the same input image. Specifically, we perform weak data augmentations to the unlabeled (or weakly labeled) image $x_{s_i}^j$ and input the image into the teacher network. We then use the output from the teacher $\{(\hat{b}_{s_i}^{jr}, \hat{p}_{s_i}^{jr})\}_{r=1}^{N_R}$ directly as pseudo-ground truth *without post-processing $f_{post}$*. To update the student, we input the *same weakly augmented image $x_{s_i}^j$* into the student and calculate the regularization loss $\mathcal{L}^{\text{regul.}}$ as follows:

$$\mathcal{L}^{\text{student}}(\theta) = \mathcal{L}_{s_1}^{\text{sup}}(\theta) + \sum_{i=2}^{N_D}[\mathcal{L}_{s_i}^{\text{unsup}}(\theta) + \beta \mathcal{L}_{s_i}^{\text{regul.}}(\theta)] \tag{8}$$

$$\begin{aligned}
\mathcal{L}_{s_i}^{\text{regul.}}(\theta) = {} & \mathcal{L}_{\text{RPN}}^{\text{cls}}(\theta, X_{s_i}, \hat{B}_{s_i}, \hat{C}_{s_i}) + \mathcal{L}_{\text{RPN}}^{\text{reg}}(\theta, X_{s_i}, \hat{B}_{s_i}, \hat{C}_{s_i}) \\
& + \mathcal{L}_{\text{RoI}}^{\text{cls}}(\theta, X_{s_i}, \hat{B}_{s_i}, \hat{C}_{s_i}) + \mathcal{L}_{\text{RoI}}^{\text{reg}}(\theta, X_{s_i}, \hat{B}_{s_i}, \hat{C}_{s_i}),
\end{aligned} \tag{9}$$

where $\hat{B}_{s_i} = \{\hat{b}_{s_i}^j\}_{j=1}^{N_{s_i}}$ and $\hat{C}_{s_i} = \{\hat{c}_{s_i}^j\}_{j=1}^{N_{s_i}}$ are the *raw pseudo-labels* from the teacher, and $\beta$ is a hyperparameter to tune the strength of the regularization.

The differences between the regularization and the traditional Mean Teacher loss in Sec. 4.3 are 1) the use of weak augmentation instead of strong augmentation, and 2) the omission of post-processing (i.e., the sharpening function of (Chen et al., 2022) in our experiments). These approaches ensure that 1) the same input is given to both the student and teacher, and 2) the raw output from the teacher is used as pseudo-labels, which encourages closer alignment between the student and teacher.

### 6.2 Connection to Prior Arts

We can regard the regularization method as a type of knowledge distillation as the student is trained to mimic the raw output from the teacher. Although the technical details are different, it has been empirically shown that knowledge distillation methods are effective on related tasks such as Single-DGOD (Wu & Deng, 2022), domain adaptive semantic segmentation (Zhang et al., 2021), UDA-OD (Cao et al., 2023; Deng et al., 2023), and semi-supervised domain adaptive object detection (where a small part of labeled target data $\mathcal{D}_t = \{(X_t, C_t)\}$ is accessible during the training (Zhou et al., 2023a)). We believe that our interpretation revealed one of the reasons knowledge-distillation methods lead to better generalization ability.

Although the technical novelty of the regularization is not our main contribution, there are technical differences from recent Mean Teacher-based methods. For example, unlike Harmonious Teacher (Deng et al., 2023), which regularizes the consistency between the classification and localization scores, our regularization

encourages consistency between the raw outputs from the teacher and student. Unlike SSDA-YOLO (Zhou et al., 2023a), we use the raw output without post-processing to produce more similar outputs, and its effectiveness is shown in Appendix A.2.1 (Table 4). More importantly, our contribution lies in providing a new interpretation of why such consistency enhances robustness to unseen domains.

## 7 Experiments

### 7.1 Dataset Details

We used the artistic style image dataset (Inoue et al., 2018), which has four domains: natural image, clipart, comic, and watercolor. The natural image domain has 16,551 images from PASCAL VOC07&12, and the other domains have 1,000, 2,000, and 2,000 images, respectively. There are six object classes (bike, bird, car, cat, dog, and person), and we removed the images that do not contain these classes.

We conducted the experiments on three patterns of domains. In the first pattern, we set the natural image domain as the labeled domain $s_1$ and set clipart and comic as the unlabeled domains $s_2, s_3$. We set watercolor as the target domain $t$. Concretely, we used the labeled trainval set of PASCAL VOC 2007&2012, the unlabeled train set of clipart, and the unlabeled train set of comic for training. We then used the test sets of clipart and comic for validation. For evaluation (testing), we used the test set of watercolor. In the second and third patterns, we set $(s_1, s_2, s_3, t) = (\text{natural}, \text{watercolor}, \text{comic}, \text{clipart})$ and $(s_1, s_2, s_3, t) = (\text{natural}, \text{watercolor}, \text{clipart}, \text{comic})$, respectively. The results on another dataset are shown in Appendix B.

### 7.2 Implementation Details

We used soft pseudo labeling proposed in (Chen et al., 2022) for the Mean Teacher learning. We used Gaussian FasterRCNN (Chen et al., 2022) as the object detector, in which the regression output is modified to use the soft labels. We used cross-entropy loss for both classification and regression losses, similar to (Chen et al., 2022). We applied the same hyperparameters as in a previous study (Chen et al., 2022) except for the number of iterations. All training (including baseline models) was done with four A100 GPUs. The parameters of the backbone network were initialized with the ResNet101 pre-trained on ImageNet. The hyperparameters $\alpha$ in Eq. (7) and $\beta$ in Eq. (8) were set to 0.9996 and 0.5 throughout the experiments, respectively. During the inference (testing) phase, we used the teacher network. Other details are given in Appendix C.

### 7.3 Baseline Methods

As the baseline, we trained the detector *Gaussian FasterRCNN* on Single-DGOD setting (i.e., supervised learning on $s_1$ in Eq. (1)). To show the effectiveness of the EMA update, we trained *Gaussian FasterRCNN + EMA* with Eqs. (6-7). *Gaussian FasterRCNN + EMA + PL* is a detector trained with the Mean Teacher learning framework in Sec. 4. *Gaussian FasterRCNN + EMA + PL + Regul.* is a detector with the Mean Teacher learning framework and the regualization in Eqs. (8-9).

To confirm the upper-bound performance, we also trained Gaussian FasterRCNN on DGOD and Oracle settings. On DGOD, the detector was trained with supervised learning using the ground-truth labels on the domains $s_1, s_2$, and $s_3$. On Oracle, the detector was trained with supervised learning on $s_1, s_2, s_3$, and the target domain $t$.

Because there is only one existing method on SS-DGOD (i.e., CDDMSL (Malakouti & Kovashka, 2023)), we also compared the above detectors with state-of-the-art methods on related task settings such as Single-DGOD and UDA-OD. It is noteworthy that existing DGOD methods such as (Lin et al., 2021; Liu et al., 2020) cannot be applied to SS-DGOD and WS-DGOD because they require labeled data from multiple source domains for training. It is important to reiterate that our goal is not to propose a new method that outperforms state-of-the-art methods. Instead, our goal is to offer novel interpretations of the Mean Teacher and demonstrate that introducing simple regularization can lead to flatter minima, resulting in better robustness to unseen domains.

Table 2: Comparisons of mAP50 on the artistic style image dataset (Inoue et al., 2018) when the target domain is watercolor. Values with * are from previous study (Li et al., 2022).

| setting | method | backbone | mAP50 | | |
|---|---|---|---|---|---|
| | | | watercolor | clipart | comic |
| Single-DGOD | CLIP-based augmentation (Vidit et al., 2023) | Res101 | 46.6 | 27.2 | **31.4** |
| Single-DGOD | Gaussian FasterRCNN | Res101 | 50.5 | 34.5 | 26.6 |
| Single-DGOD | Gaussian FasterRCNN + EMA | Res101 | **55.5** | **38.0** | 29.0 |
| SS-DGOD | CDDMSL (Malakouti & Kovashka, 2023) | Res50 (RegionCLIP) | 46.1 | 39.1 | **38.3** |
| SS-DGOD | CDDMSL (Malakouti & Kovashka, 2023) | Res101 | 41.3 | 26.0 | 28.8 |
| SS-DGOD | Gaussian FasterRCNN + EMA + PL | Res101 | 56.6 | 39.8 | 30.1 |
| SS-DGOD | Gaussian FasterRCNN + EMA + PL + Regul. | Res101 | **58.2** | **43.3** | 32.2 |
| WS-DGOD | Gaussian FasterRCNN + EMA + PL | Res101 | 59.7 | 44.2 | 39.9 |
| WS-DGOD | Gaussian FasterRCNN + EMA + PL + Regul. | Res101 | **62.9** | **46.2** | **40.2** |
| DGOD | Gaussian FasterRCNN | Res101 | 62.6 | 47.1 | 45.2 |
| Oracle | Gaussian FasterRCNN | Res101 | 62.2 | 48.2 | 48.6 |
| UDA-OD | Gaussian FasterRCNN + EMA + PL (Chen et al., 2022) | Res101 | 54.9 | 43.4 | 27.0 |
| UDA-OD | Gaussian FasterRCNN + EMA + PL + Regul. | Res101 | 58.8 | 45.4 | 32.7 |
| UDA-OD | SCL* (Shen et al., 2019) | Res101 | 55.2 | - | - |
| UDA-OD | SWDA* (Saito et al., 2019) | Res101 | 53.3 | - | - |
| UDA-OD | UMT* (Deng et al., 2021) | Res101 | 58.1 | - | - |
| UDA-OD | AT* (Li et al., 2022) | Res101 | 59.9 | - | - |

Although there are previous semi-supervised domain generalization methods proposed for image classification, extending them to object detection is non-trivial and they cannot be directly applied to SS-DGOD. Therefore, we compared our method with CDDMSL in our experiments, which is the previous work specifically tailored for SS-DGOD. Regarding weakly-supervised domain generalization, there are no previous methods, even for the image classification task, because weak labels cannot be defined in image classification.

### 7.4 Comparisons with Other Methods

Table 2 shows the results on the artistic image style dataset. We evaluated with the mean average precision (mAP50) when the IoU threshold was 0.5. EMA increased the mAP of *Gaussian FasterRCNN* from (50.5, 34.5, 26.6) to (55.5, 38.0, 29.0), and this was further boosted to (56.6, 39.8. 30.1) with pseudo labeling (PL). We observed additional improvement to (58.2, 43.3, 32.2) with the regularization. The regularization improved the performance not only on SS-DGOD but also on WS-DGOD. The detectors trained on WS-DGOD performed better than those on SS-DGOD because WS-DGOD can generate more accurate pseudo labels by the refinement in Eq. (2). Those results are comparable to those of the detectors trained on DGOD and Oracle.

For fair comparisons, we trained CDDMSL with Res101 backbone pre-trained on ImageNet. However, its performance significantly degraded, as reported in a previous study (Malakouti & Kovashka, 2023), because it requires language-guided training, and initializing the model with RegionCLIP is crucial to achieve good performance.

The detectors trained on SS-DGOD and WS-DGOD also performed comparably to or better than those on UDA-OD, although we did not use the target domain data during the training. Furthermore, the regularization can be directly applied to UDA-OD as well as SS-DGOD and WS-DGOD, and we also observed significant performance improvement by the regularization on UDA-OD.

### 7.5 Analysis of Flatness

To evaluate the flatness of the detectors in parameter space, following previous studies (Izmailov et al., 2018) and (Cha et al., 2021), we computed the change in loss values when we perturb the parameters. Specifically, we sampled a random direction vector $d$ on a unit sphere, perturbed the parameters $(\theta' = \theta + d\gamma)$ with a radius $\gamma$, and computed the average change over ten samples, i.e., $\mathcal{F}^\gamma(\theta) = \mathbb{E}_{\theta'}|\mathcal{E}(\theta') - \mathcal{E}(\theta)|$. The lower the change is, the flatter the parameters.

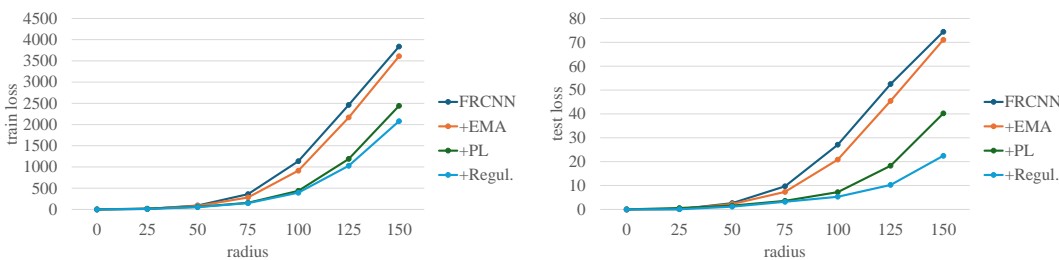

Figure 7: Left and right plots compare average training and test flatness, respectively.

Fig. 7 shows the $\mathcal{F}^{\gamma}(\theta)$ of the training loss $\mathcal{E}(\theta) = \sum_i \mathcal{L}_{s_i}^{sup}(\theta)$ and the test loss $\mathcal{E}(\theta) = \mathcal{L}_t^{sup}(\theta)$. The training domains were $(s_1, s_2, s_3)$=(natural, watercolor, comic), and the test domain was clipart. We can see that EMA, PL, and the regularization lowered the changes in the losses on both the training domains and test domain. In other words, each contributed to falling into flatter minima.

## 8    Conclusion and Limitation

We tackled two problem settings called semi-supervised domain generalizable object detection (SS-DGOD) and weakly-supervised DGOD (WS-DGOD) to train object detectors that can generalize to unseen domains. We showed that the object detectors can be effectively trained on the two settings with the same Mean Teacher learning framework. We also provided the interpretations of why the detectors trained with the Mean Teacher framework become robust to the unseen domains in terms of the flatness in the parameter space. Based on the interpretations, we introduced a regularization method to lead to flatter minima, which makes the loss value of the student similar to that of the teacher. The experiments showed that the detectors trained with the Mean Teacher learning framework and the regularization performed significantly better than the baseline methods. Because Mean Teacher has been used across various tasks, our novel interpretation of why Mean Teacher becomes robust to unknown domains is likely to have a broad impact across a wide range of tasks.

Each of the Mean Teacher and flat minima theory (the relationship between the model generalization ability and flatness of the solution in loss landscapes) are already established concepts, separately. Different from them, our main contribution lies in the novel finding that the Mean Teacher framework leads to the flat minima and the interpretation of the reasoning behind it. On the basis of the flat minima theory and our novel finding, we explained the reason of the detectors trained with the Mean Teacher learning framework achieve robustness to unseen test domains.

The limitation is that the assumption in Sec. 5.4 does not always hold. Specifically, it is not always guaranteed that the pseudo labels from the teacher are accurate enough to approximate $\mathcal{L}_{s_i}^{unsup}$ with $\mathcal{L}_{s_i}^{sup}$. Nevertheless, we empirically showed that the Mean Teacher and the regularization lead to flatter minima in practice. There are two primary reasons for this observation. First, when considering each domain independently, the assumption always holds in the labeled domain $s_1$, as labeled data is available, ensuring that $\mathcal{L}_{s_1}^{unsup} = \mathcal{L}_{s_1}^{sup}$. Second, the assumption is only necessary to explain how the Mean Teacher achieves flat minima in the *empirical risk* (i.e., the sum of the supervised losses $\mathcal{E}_{\mathrm{ER}}(\theta) = \sum_{i=1}^{N_D} \mathcal{L}_{s_i}^{\mathrm{sup}}(\theta)$). Even if this assumption does not hold, we can similarly explain that the Mean Teacher reaches flat minima in the sum of supervised and unsupervised losses in Eq. (3). We believe that achieving flat minima in Eq. (3) still positively affects robustness against unseen domains. Further analysis of failure cases is left for future work.

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

# Appendix

## A    More Analysis

### A.1    How Sensitive to Hyperparameter $\beta$?

Table 3 shows the performance when the hyperparameter $\beta$ in Eq. (8) (i.e. strength of the regularization) was changed from 0 to 1. By adding the regularization, the performance was constantly improved from the detector without regularization (i.e., $\beta = 0$).

Table 3: mAP50 with various $\beta$ on the artistic style image dataset (Inoue et al., 2018).

| setting | method | $\beta$ | mAP50 |
| --- | --- | --- | --- |
| | | | clipart |
| SS-DGOD | Gaussian FasterRCNN + EMA + PL | 0.0 | 39.8 |
| SS-DGOD | Gaussian FasterRCNN + EMA + PL + Regul. | 0.25 | 40.7 |
| SS-DGOD | Gaussian FasterRCNN + EMA + PL + Regul. | 0.5 | **43.3** |
| SS-DGOD | Gaussian FasterRCNN + EMA + PL + Regul. | 0.75 | 42.1 |
| SS-DGOD | Gaussian FasterRCNN + EMA + PL + Regul. | 1.0 | 42.5 |

### A.2    Importance of Encouraging Consistency

#### A.2.1    Comparison of Regularization with and without Post-processing

In the regularization described in Sec. 6.1, we use the raw outputs from the teacher *without post-processing* to train the student so that the outputs from the two networks are similar. To validate the claim, we compare the performance with and without post-processing (i.e., sharpening function (Chen et al., 2022)) in the regularization in Eq. (9). Table 4 shows that the performance drops when we perform the post-processing. We observe that using raw outputs is important to obtain better performance.

Table 4: mAP50 with and without post-processing on the artistic style image dataset (Inoue et al., 2018).

| setting | method | post process | mAP50 |
| --- | --- | --- | --- |
| | | | clipart |
| SS-DGOD | Gaussian FasterRCNN + EMA + PL + Regul. | | **43.3** |
| SS-DGOD | Gaussian FasterRCNN + EMA + PL + Regul. | ✓ | 39.4 |

#### A.2.2    Importance of Consistent Augmentation between Teacher and Student

In the regularization described in Sec. 6.1, we input *weakly*-augmented images to the student (i.e., same input as the teacher) in order to encourage the consistency between the outputs from the teacher and student. In this section, as shown in Fig. 8, we compare the weak and strong augmentation for the student in the regularization whereas weakly-augmented images were always input to the teacher. Table 5 shows that the weak augmentation obtains better performance than the strong augmentation, which implies the consistency between the outputs from the teacher and student using the same inputs leads to better performance.

Table 5: Comparison of mAP50 between strong and weak augmentation in the regularization on the artistic style image dataset (Inoue et al., 2018).

| setting | method | augmentation | mAP50 |
| --- | --- | --- | --- |
| | | | clipart |
| SS-DGOD | Gaussian FasterRCNN + EMA + PL + Regul. | weak | **43.3** |
| SS-DGOD | Gaussian FasterRCNN + EMA + PL + Regul. | strong | 42.5 |

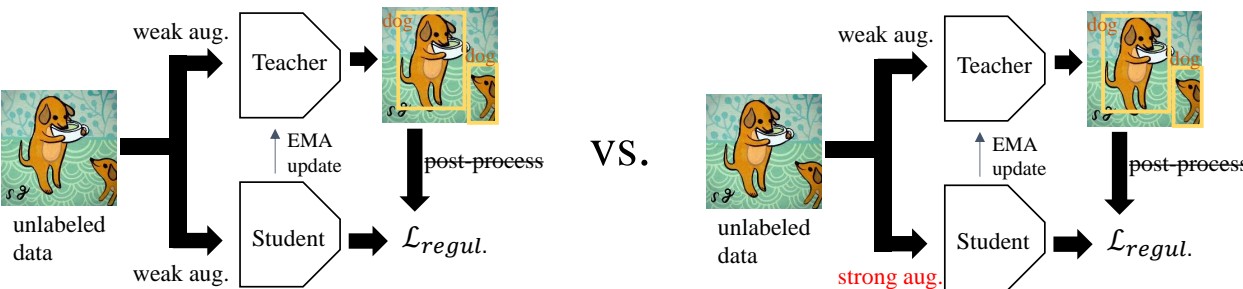

Figure 8: Comparisons between weak and strong augmentation for the student in the regularization.

## A.3 Why is Only Weak Augmentation Used in the Regularization?

One may think why only weak augmentation is used in the regualization in Fig. 6, and what is the performance of randomly using strong and weak augmentation? To answer this question, we evaluated the performance when randomly using strong and weak augmentation as shown in the right side of Fig. 9. In this setting, the strong and weak augmentation was randomly chosen with a probability of 0.5 at each iteration. When the strong augmentation was chosen, the same strongly augmented image was input into both the student and teacher networks. Then, the raw output from the teacher without post-processing was used to calculate the regularization loss for the students in Eqs. (8) and (9) to encourage consistency between the outputs from the student and teacher. As shown in Table 6, only weak augmentation obtained better performance. We think it is because inputting strongly augmented images into the teacher can make noisy pseudo-labels.

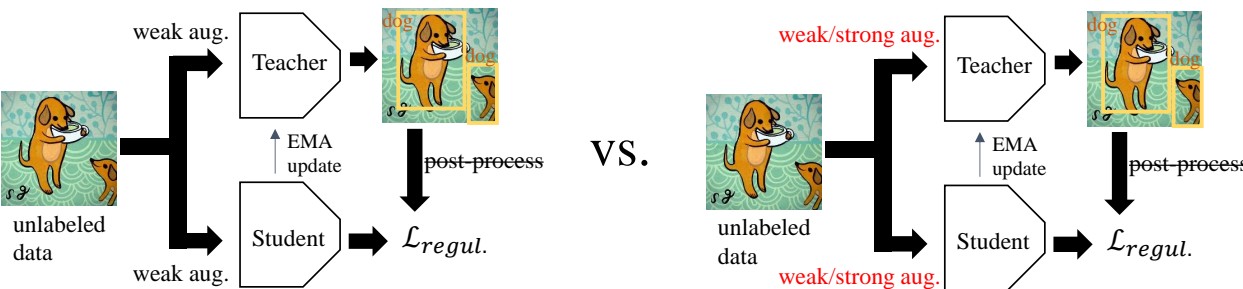

Figure 9: Comparisons between weak and random (weak/strong) augmentation in the regularization.

Table 6: Comparison of mAP50 between weak and weak/strong (random) augmentation in the regularization on the artistic style image dataset.

| setting | method | augmentation | mAP50 |
|---------|--------|--------------|-------|
| | | | clipart |
| SS-DGOD | Gaussian FasterRCNN + EMA + PL | N/A | 39.8 |
| SS-DGOD | Gaussian FasterRCNN + EMA + PL + Regul. | weak | **43.3** |
| SS-DGOD | Gaussian FasterRCNN + EMA + PL + Regul. | weak/strong (random) | 41.1 |

## A.4 Does the regularization make it difficult or slow to train the model?

One may be concerned whether the regularization makes the training difficult or slow because the regularization in Fig. 6 encourages the teacher and student to produce similar predictions. To address the concern, in Fig. 10, we show the mAP on the validation set during training with and without regularization. The regularization does not make the training process more difficult or slower. On the contrary, the regularization helps alleviate overfitting (i.e., less decrease in the validation mAP), stabilizing the training.

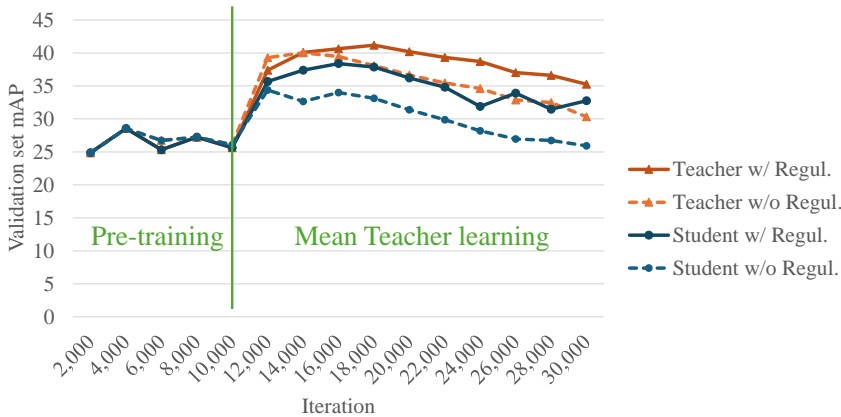

Figure 10: mAP on the validation set during the training.

## A.5 Class-wise Average Precision

Table 7, 8, and 9 show average precision (AP50) at each class when the target domain is watercolor, clipart, and comic, respectively. We can see that the regularization improved the performance on many classes.

In Table 7, *Gaussian FasterRCNN + EMA + PL + Regul.* trained on the WS-DGOD setting slightly outperforms *Gaussian FasterRCNN* trained on the DGOD and Oracle settings. The potential reason is that DGOD and Oracle were trapped in sharp local minima due to simple supervised learning (i.e., ERM), while *Gaussian FasterRCNN + EMA + PL + Regul.* reached flat minima. It has been shown that even when both the train and test sets are from the same domain, there is a slight shift between the train loss and test loss, and falling into a sharp valley can decrease performance (Izmailov et al., 2018). Fig. 11 shows the comparison of the flatness similar to that in Sec. 7.5. We observe that *Gaussian FasterRCNN + EMA + PL + Regul.* trained on the WS-DGOD achieved a flatter solution than *Gaussian FasterRCNN* trained on the DGOD and Oracle settings.

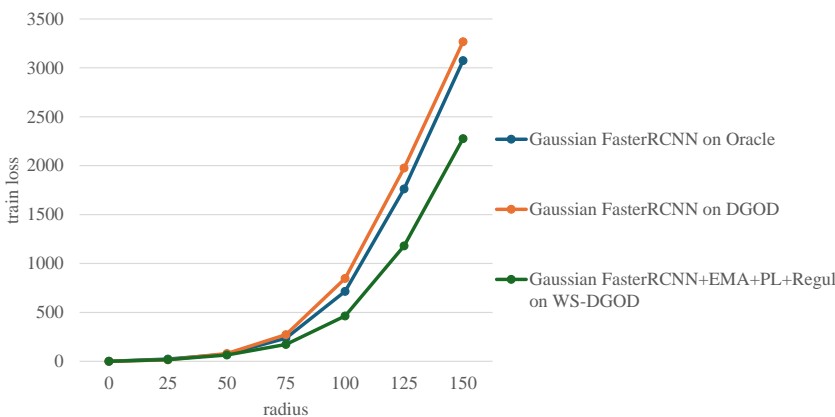

Figure 11: Comparison of flatness of train loss.

## A.6 Qualitative Results

Figs. 12, 13, and 14 show the qualitative comparison on watercolor, clipart, and comic, respectively. We observe that false negative detection of the baseline model was drastically improved.

Table 7: Comparisons of AP50 at each class on watercolor of the artistic style image dataset (Inoue et al., 2018). The values of * are from (Li et al., 2022).

| setting | method | bicycle | bird | cat | car | dog | person | mAP |
|---|---|---|---|---|---|---|---|---|
| Single-DGOD | CLIP-based augmentation (Vidit et al., 2023) | 74.8 | 37.3 | **36.8** | 40.7 | 29.2 | 59.9 | 46.4 |
| Single-DGOD | Gaussian FasterRCNN | **90.4** | 47.9 | 30.3 | 46.7 | 28.7 | 59.2 | 50.5 |
| Single-DGOD | Gaussian FasterRCNN + EMA | 86.2 | **54.3** | 35.3 | **53.5** | **34.5** | **69.0** | **55.5** |
| SS-DGOD | CDDMSL (Malakouti & Kovashka, 2023) (RegionCLIP) | 66.3 | 50.6 | 34.5 | 49.2 | 20.1 | 56.0 | 46.1 |
| SS-DGOD | CDDMSL (Malakouti & Kovashka, 2023) (Res101) | 75.5 | 36.1 | 23.9 | 40.7 | 19.7 | 52.0 | 41.3 |
| SS-DGOD | Gaussian FasterRCNN + EMA + PL | **87.4** | **54.6** | 40.0 | 51.9 | 32.4 | 73.1 | 56.6 |
| SS-DGOD | Gaussian FasterRCNN + EMA + PL + Regul. | 87.2 | 52.3 | **44.7** | **53.2** | **36.8** | **75.3** | **58.2** |
| WS-DGOD | Gaussian FasterRCNN + EMA + PL | 90.3 | 55.8 | 49.3 | 49.9 | 37.5 | 75.4 | 59.7 |
| WS-DGOD | Gaussian FasterRCNN + EMA + PL + Regul. | **95.8** | **59.9** | **51.5** | **53.3** | **40.2** | **76.7** | **62.9** |
| DGOD | Gaussian FasterRCNN | 84.8 | 57.8 | 51.0 | 50.8 | 51.8 | 79.3 | 62.6 |
| Oracle | Gaussian FasterRCNN | 90.9 | 59.9 | 44.2 | 53.1 | 46.7 | 78.3 | 62.2 |
| UDA-OD | Gaussian FasterRCNN + EMA + PL (Chen et al., 2022) | 77.7 | 46.5 | 40.4 | 50.1 | 39.7 | 75.0 | 54.9 |
| UDA-OD | Gaussian FasterRCNN + EMA + PL + Regul. | 82.8 | 51.4 | 43.2 | 59.3 | 39.0 | 77.0 | 58.8 |
| UDA-OD | SCL* (Shen et al., 2019) | 82.2 | 55.1 | 51.8 | 39.6 | 38.4 | 64.0 | 55.2 |
| UDA-OD | SWDA* (Saito et al., 2019) | 82.3 | 55.9 | 46.5 | 32.7 | 35.5 | 66.7 | 53.3 |
| UDA-OD | UMT* (Deng et al., 2021) | 88.2 | 55.3 | 51.7 | 39.8 | 43.6 | 69.9 | 58.1 |
| UDA-OD | AT* (Li et al., 2022) | 93.6 | 56.1 | 58.9 | 37.3 | 39.6 | 73.8 | 59.9 |

Table 8: Comparisons of AP50 at each class on clipart of the artistic style image dataset (Inoue et al., 2018).

| setting | method | bicycle | bird | cat | car | dog | person | mAP |
|---|---|---|---|---|---|---|---|---|
| Single-DGOD | CLIP-based augmentation (Vidit et al., 2023) | 36.5 | 22.5 | **20.1** | 25.0 | 8.8 | 50.4 | 27.2 |
| Single-DGOD | Gaussian FasterRCNN | 69.5 | 25.1 | 5.7 | **39.4** | 17.3 | 49.9 | 34.5 |
| Single-DGOD | Gaussian FasterRCNN + EMA | **87.6** | **29.3** | 5.5 | 30.1 | **18.3** | **57.2** | **38.0** |
| SS-DGOD | CDDMSL (Malakouti & Kovashka, 2023) (RegionCLIP) | 51.0 | **33.3** | 26.5 | 45.2 | 14.6 | 63.8 | 39.1 |
| SS-DGOD | CDDMSL (Malakouti & Kovashka, 2023) (Res101) | 41.6 | 19.2 | 5.5 | 26.7 | 12.3 | 50.9 | 26.0 |
| SS-DGOD | Gaussian FasterRCNN + EMA + PL | 75.8 | 31.2 | 9.4 | 33.1 | 20.4 | **69.1** | 39.8 |
| SS-DGOD | Gaussian FasterRCNN + EMA + PL + Regul. | **79.3** | 32.5 | 11.6 | 40.9 | **26.3** | 69.0 | **43.3** |
| WS-DGOD | Gaussian FasterRCNN + EMA + PL | 80.3 | **33.3** | 11.1 | **44.5** | **23.2** | **72.6** | 44.2 |
| WS-DGOD | Gaussian FasterRCNN + EMA + PL + Regul. | **84.8** | 33.2 | **23.8** | 43.0 | 22.1 | 70.1 | **46.2** |
| DGOD | Gaussian FasterRCNN | 76.0 | 34.8 | 18.8 | 38.3 | 36.9 | 77.6 | 47.1 |
| Oracle | Gaussian FasterRCNN | 70.4 | 38.8 | 26.1 | 52.9 | 27.5 | 73.4 | 48.2 |
| UDA-OD | Gaussian FasterRCNN + EMA + PL (Chen et al., 2022) | 79.9 | 33.5 | 6.5 | 53.1 | 23.7 | 65.2 | 43.6 |
| UDA-OD | Gaussian FasterRCNN + EMA + PL + Regul. | 72.4 | 35.4 | 16.0 | 57.2 | 19.7 | 71.5 | 45.4 |

## A.7 When the Number of Unlabeled Domain is One

In Sec. 7, we conducted the experiments under the setting of one labeled domain and multiple unlabeled domains (e.g., $(s_1, s_2, s_3, t)$= (natural, clipart, comic, watercolor)). In this section, we conducted the experiments with one labeled domain and one unlabeled domain, which are the same settings as the CDDSL paper (Malakouti & Kovashka, 2023): (s1, s2, t) = (natural, comic, watercolor) and (natural, comic, clipart). Table 10 shows the superior performance to CDDMSL even on these settings.

## A.8 Different Backbone

To validate the generalization ability of the regularization, we conducted the experiments with another detector that has a significantly different network design. Specifically, we used a Transformer-based backbone (Swin-T) (Liu et al., 2021) with the feature pyramid network (Lin et al., 2017), although the detection head was not changed. Table 11 shows the results. We can see that the regularization improves the performance, which validates its generalization ability.

## A.9 Comparison and Combination with Another Existing Trick to Improve Flat Minima

Table12 shows the comparison with another existing method to find flat minima called Sharpness-Aware Minimization (SAM) (Foret et al., 2021). By comparing *Gaussian FasterRCNN + EMA + PL + SAM* and *Gaussian FasterRCNN + EMA + PL + Regul.*, our regularization outperforms the SAM. In addition, since the SAM is an optimizer and can be used instead of SGD, it is compatible with our regularization. We can see that *Gaussian FasterRCNN + EMA + PL + Regul. + SAM* achieved the best performance.

Table 9: Comparisons of AP50 at each class on comic of the artistic style image dataset (Inoue et al., 2018).

| setting | method | bicycle | bird | cat | car | dog | person | mAP |
|---------|--------|---------|------|-----|-----|-----|--------|-----|
| Single-DGOD | CLIP-based augmentation (Vidit et al., 2023) | 29.0 | **18.6** | **27.6** | 32.7 | **28.4** | **52.2** | **31.4** |
| Single-DGOD | Gaussian FasterRCNN | 45.0 | 10.8 | 9.5 | **33.8** | 17.5 | 43.0 | 26.6 |
| Single-DGOD | Gaussian FasterRCNN + EMA | **50.0** | 15.0 | 11.2 | 26.8 | 22.4 | 48.3 | 29.0 |
| SS-DGOD | CDDMSL (Malakouti & Kovashka, 2023) (RegionCLIP) | 41.8 | **27.8** | **23.5** | 44.2 | **34.8** | 57.8 | **38.3** |
| SS-DGOD | CDDMSL (Malakouti & Kovashka, 2023) (Res101) | **44.3** | 12.2 | 13.7 | 30.5 | 19.7 | 52.1 | 28.8 |
| SS-DGOD | Gaussian FasterRCNN + EMA + PL | 41.5 | 14.5 | 11.4 | 24.5 | 27.3 | **61.1** | 30.1 |
| SS-DGOD | Gaussian FasterRCNN + EMA + PL + Regul. | 42.3 | 15.6 | 15.9 | 31.5 | 30.2 | 57.8 | 32.2 |
| WS-DGOD | Gaussian FasterRCNN + EMA + PL | 53.7 | 23.1 | 19.9 | **44.3** | **33.7** | **64.5** | 39.9 |
| WS-DGOD | Gaussian FasterRCNN + EMA + PL + Regul. | **54.2** | **23.2** | **23.8** | 44.1 | 31.5 | 64.2 | **40.2** |
| DGOD | Gaussian FasterRCNN | 54.6 | 29.5 | 33.5 | 38.9 | 43.2 | 71.4 | 45.2 |
| Oracle | Gaussian FasterRCNN | 55.7 | 29.0 | 44.5 | 46.3 | 45.1 | 71.3 | 48.6 |
| UDA-OD | Gaussian FasterRCNN + EMA + PL (Chen et al., 2022) | 42.2 | 13.6 | 10.8 | 16.6 | 19.3 | 59.5 | 27.0 |
| UDA-OD | Gaussian FasterRCNN + EMA + PL + Regul. | 46.3 | 14.4 | 20.3 | 28.8 | 23.5 | 62.6 | 32.7 |

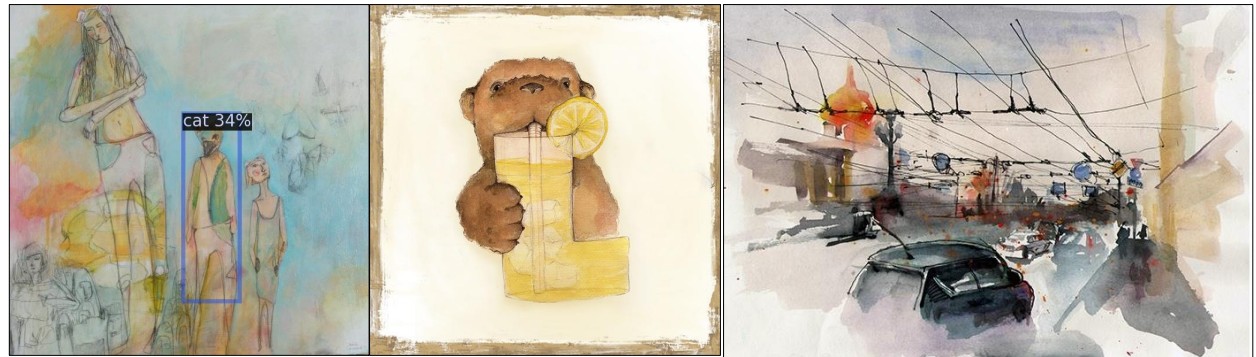

(a) Gaussian FasterRCNN trained on Single-DGOD setting (i.e., trained with labeled data on PASCAL VOC07&12).

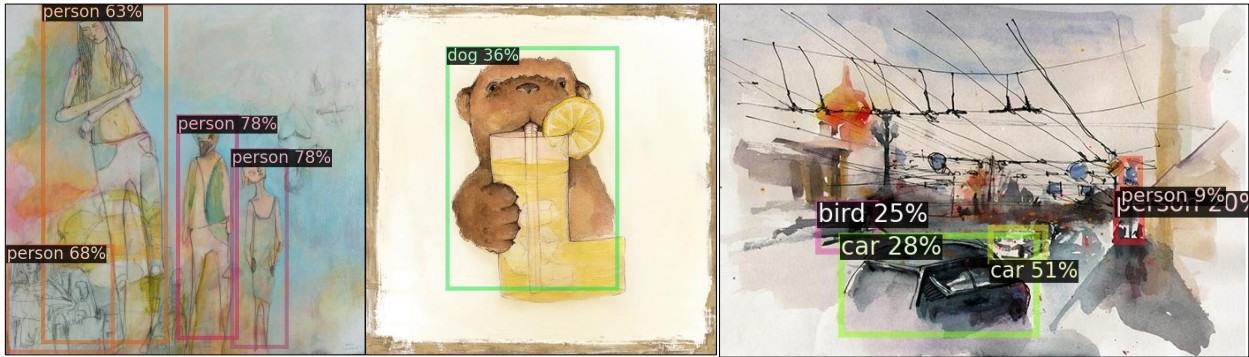

(b) Gaussian FasterRCNN + EMA + PL + Regul. trained on SS-DGOD setting (i.e., trained with labeled data on PASCAL VOC07&12 and unlabeled data on clipart and comic).

Figure 12: Qualitative comparisons on watertcolor.

Table 10: Comparisons of mAP50 on the artistic style image dataset when $(s_1, s_2, t)=$ (natural, comic, watercolor) and (natural, comic, clipart). Values with * are from previous study (Malakouti & Kovashka, 2023).

| setting | method | backbone | mAP50 | |
|---------|--------|----------|-------|---|
| | | | watercolor | clipart |
| SS-DGOD | CDDMSL* (Malakouti & Kovashka, 2023) | Res50 (RegionCLIP) | 49.4 | 39.8 |
| SS-DGOD | Gaussian FasterRCNN + EMA + PL | Res101 | 55.2 | 38.4 |
| SS-DGOD | Gaussian FasterRCNN + EMA + PL + Regul. | Res101 | **56.5** | **40.1** |

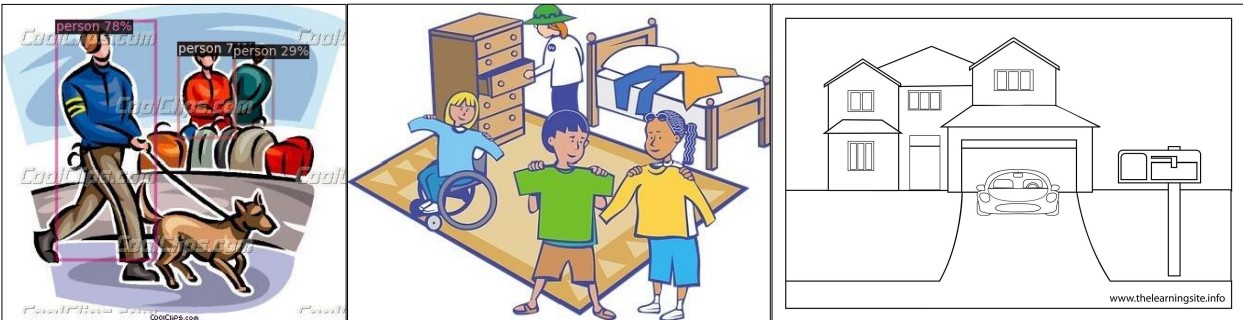

(a) Gaussian FasterRCNN trained on Single-DGOD setting (i.e., trained with labeled data on PASCAL VOC07&12).

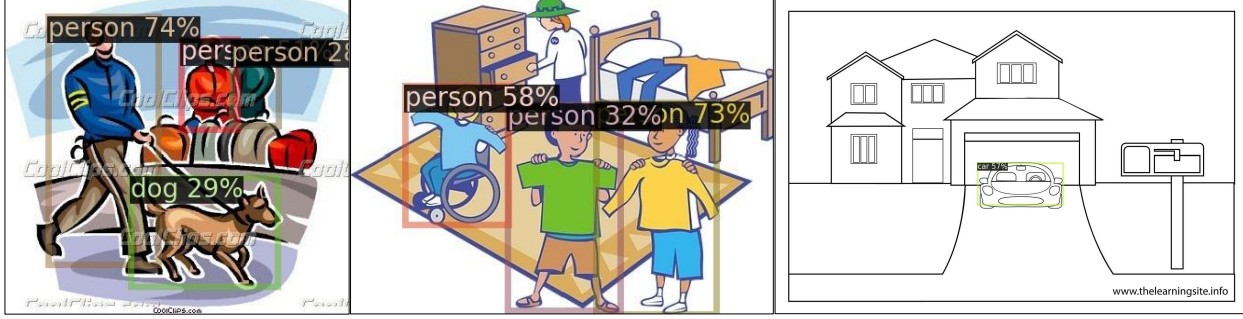

(b) Gaussian FasterRCNN + EMA + PL + Regul. trained on SS-DGOD setting (i.e., trained with labeled data on PASCAL VOC07&12 and unlabeled data on watercolor and comic).

Figure 13: Qualitative comparisons on clipart.

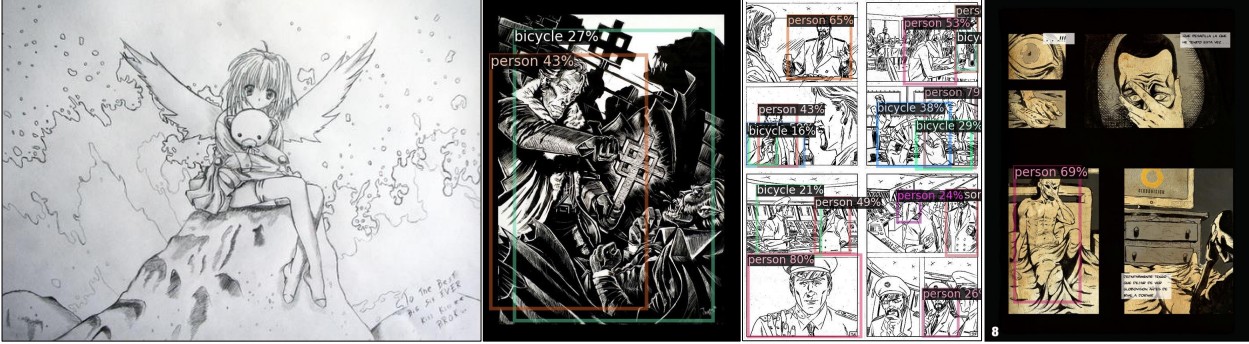

(a) Gaussian FasterRCNN trained on Single-DGOD setting (i.e., trained with labeled data on PASCAL VOC07&12).

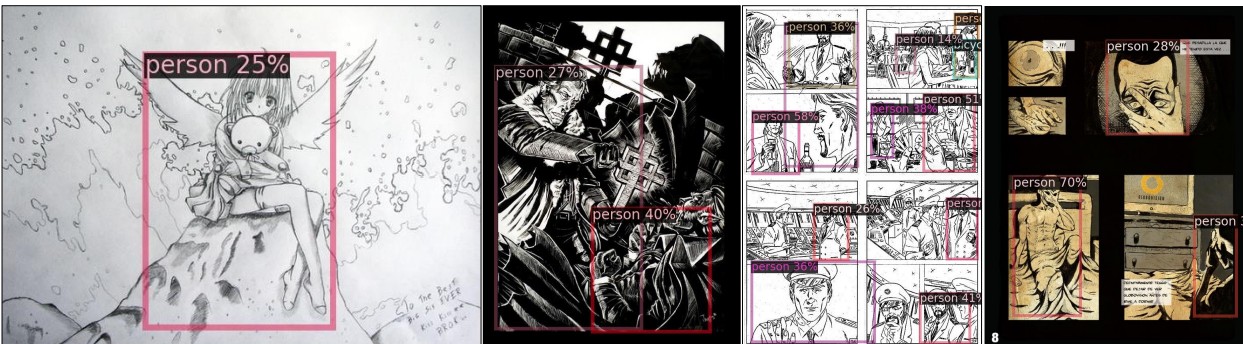

(b) Gaussian FasterRCNN + EMA + PL + Regul. trained on SS-DGOD setting (i.e., trained with labeled data on PASCAL VOC07&12 and unlabeled data on clipart and comic).

Figure 14: Qualitative comparisons on comic.

Table 11: Comparison of mAP50 with and without regularization using Swin-T + FPN backbone on the artistic style image dataset (Inoue et al., 2018).

| setting | method | backbone | mAP50 |
| --- | --- | --- | --- |
| | | | watercolor |
| SS-DGOD | Gaussian FasterRCNN + EMA + PL | Swin-T + FPN | 53.9 |
| SS-DGOD | Gaussian FasterRCNN + EMA + PL + Regul. | Swin-T + FPN | **55.0** |

Table 12: Comparison and combination with SAM on the artistic style image dataset (Inoue et al., 2018).

| setting | method | backbone | mAP50 |
| --- | --- | --- | --- |
| | | | watercolor |
| Single-DGOD | Gaussian FasterRCNN | Res101 | 50.5 |
| Single-DGOD | Gaussian FasterRCNN + SAM | Res101 | 54.1 |
| Single-DGOD | Gaussian FasterRCNN + EMA | Res101 | 55.5 |
| SS-DGOD | Gaussian FasterRCNN + EMA + PL | Res101 | 56.6 |
| SS-DGOD | Gaussian FasterRCNN + EMA + PL + SAM | Res101 | 57.5 |
| SS-DGOD | Gaussian FasterRCNN + EMA + PL + Regul. | Res101 | 58.2 |
| SS-DGOD | Gaussian FasterRCNN + EMA + PL + Regul. + SAM | Res101 | **59.5** |

# B    Results on Car-mounted Camera Dataset (Wu & Deng, 2022)

## B.1    Dataset Details

The car-mounted camera dataset is a recently developed dataset in (Wu & Deng, 2022) for Single-DGOD or DGOD, where the images were selected from the standard datasets such as Cityscapes (Cordts et al., 2016), FoggyCityscapes (Sakaridis et al., 2018), BDD-100k (Yu et al., 2020), and AdverseWeather (Hassaballah et al., 2020). The domains were clearly redefined based on the weather and time differences: daytime-sunny, night-sunny, daytime-foggy, dusk-rainy, and night-rainy. The number of images for each domain is 27,708, 18,310, 2,642, 3,501, and 2,494, respectively. We used daytime-sunny as the labeled domain $s_1$ and used night-sunny and daytime-foggy as the unlabeled (or weakly-labeled) domains $s_2, s_3$. We used each of the remaining domains (dusk-rainy and night-rainy) as the target domain. Because the train/val/test split is not publicly available for daytime-sunny, dusk-rainy, and night-rainy, we used all images of daytime-sunny, the trainval set of night-sunny, and the trainval set of daytime-foggy for training. We then used the test set of night-sunny and the test set of daytime-foggy for validation. We used all images of dusk-rainy and night-rainy for evaluation (testing). There are seven object classes: bus, bike, car, motor, person, rider, and truck.

There are two reasons for using this dataset for evaluation. One is that the images in this dataset were selected from the standard datasets, and the other is that the domains were clearly redefined based on the weather and time differences as described above. In the setting of the previous work (Malakouti & Kovashka, 2023), i.e., $(s_1, s_2, t) = $ (Cityscapes, FoggyCityscapes, BDD100k), the differences between domains are ambiguous. This is because Cityscape primarily assumes clear/medium daytime weather, Foggycityscape assumes foggy weather, while BDD100K includes various times of day and weather conditions. Therefore, instead, we used the car-mounted camera dataset.

## B.2    Comparisons with Other Methods

Table 13: Comparisons of mAP50 on the car-mounted camera dataset (Wu & Deng, 2022). The values of * and ** were from (Wu & Deng, 2022) and (Vidit et al., 2023), respectively.

| setting | method | backbone | mAP50 | |
|---|---|---|---|---|
| | | | dusk-rainy | night-rainy |
| Single-DGOD | FasterRCNN* | Res101 | 26.6 | 14.5 |
| Single-DGOD | CDSD* (Wu & Deng, 2022) | Res101 | 28.2 | 16.6 |
| Single-DGOD | CLIP-based augmentation**(Vidit et al., 2023) | Res101 | 32.3 | 18.7 |
| Single-DGOD | Gaussian FasterRCNN | Res101 | 25.3 | 13.3 |
| Single-DGOD | Gaussian FasterRCNN + EMA | Res101 | **36.0** | **19.0** |
| SS-DGOD | Gaussian FasterRCNN + EMA + PL | Res101 | 30.3 | 21.3 |
| SS-DGOD | Gaussian FasterRCNN + EMA + PL + Regul. | Res101 | **31.2** | **21.9** |
| WS-DGOD | Gaussian FasterRCNN + EMA + PL | Res101 | 30.5 | 22.5 |
| WS-DGOD | Gaussian FasterRCNN + EMA + PL + Regul. | Res101 | **32.5** | **23.1** |
| DGOD | Gaussian FasterRCNN | Res101 | 28.4 | 21.2 |

Table 13 shows the results on the car-mounted camera dataset. Each of EMA, PL, and the regularization improved the performance on both target domains except that PL degraded the performance on dusk-rainy. We will investigate the cause of the performance drop in our future work.

The mAP50 of the detector with the regularization is boosted to (32.5, 23.1) on WS-DGOD. This result exceeds (32.3, 18.7), which is the result of CLIP-based augmentation (Vidit et al., 2023) proposed for Single-DGOD. Also, this result is better than those of the models trained with supervised learning on the three domains (DGOD).

## B.3    Analysis of Flatness

Fig. 15 shows the average change of the training loss at each domain when perturbing the parameters ($\mathcal{F}^\gamma(\theta) = \mathbb{E}_{\theta'}|\mathcal{E}(\theta') - \mathcal{E}(\theta)|$ described in Sec. 7.5), and Fig. 16 shows those of the test loss. Each of EMA, PL,

and the regularization lowered the changes in the losses at every domain when the radius is 125 or smaller although EMA lowered the changes the most when the radius is extremely large ($> 125$). In other words, each contributed to falling into flatter minima with a sufficiently large radius.

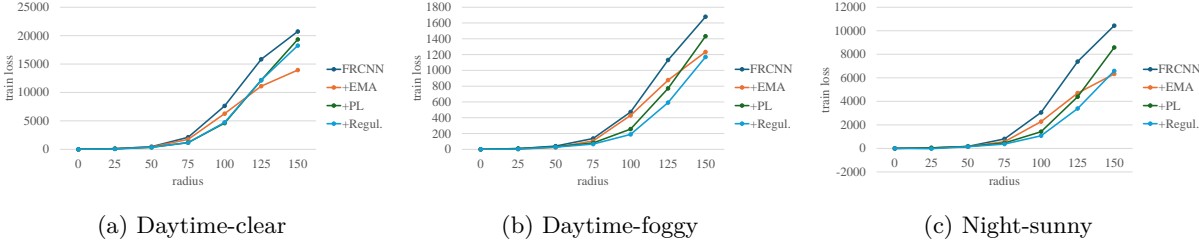

(a) Daytime-clear      (b) Daytime-foggy      (c) Night-sunny

Figure 15: Average training flatness at each training domain.

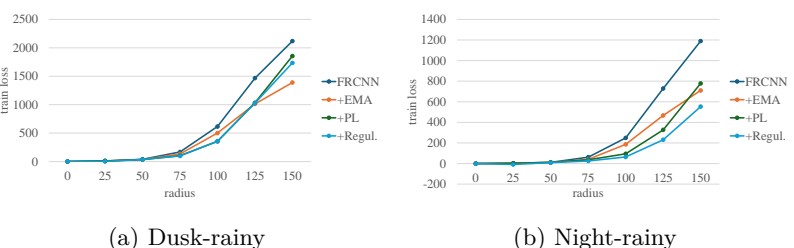

(a) Dusk-rainy      (b) Night-rainy

Figure 16: Average test flatness at each target domain.

## B.4 Class-wise Average Precision

Tables 14 and 15 show class-wise average precision on dusk-rainy and night-rainy domains, respectively. We can see that each of EMA, PL, and regularization contributes to improving the performance on many classes except the performance drop by PL on dusk-rainy.

Table 14: Comparisons of AP50 at each class on dusk-rainy of the car-mounted camera dataset (Wu & Deng, 2022). The values of * and ** are from (Wu & Deng, 2022) and (Vidit et al., 2023), respectively.

| setting | method | bus | bike | car | motor | person | rider | truck | mAP |
|---|---|---|---|---|---|---|---|---|---|
| Single-DGOD | FasterRCNN* | 36.8 | 15.8 | 50.1 | 12.8 | 18.9 | 12.4 | 39.5 | 26.6 |
| Single-DGOD | CDSD* (Wu & Deng, 2022) | 37.1 | 19.6 | 50.9 | 13.4 | 19.7 | 16.3 | 40.7 | 28.2 |
| Single-DGOD | CLIP-based augmentation** (Vidit et al., 2023) | 37.8 | 22.8 | 60.7 | **16.8** | 26.8 | 18.7 | 42.4 | 32.3 |
| Single-DGOD | Gaussian FasterRCNN | 33.9 | 14.9 | 53.6 | 4.2 | 17.4 | 13.6 | 39.2 | 25.3 |
| Single-DGOD | Gaussian FasterRCNN + EMA | **46.3** | **24.9** | **65.9** | 11.9 | **29.1** | **23.7** | **50.0** | **36.0** |
| SS-DGOD | Gaussian FasterRCNN + EMA + PL | 40.0 | 17.3 | 61.0 | **8.0** | **23.6** | 17.1 | 45.1 | 30.3 |
| SS-DGOD | Gaussian FasterRCNN + EMA + PL + Regul. | **40.8** | **20.1** | **61.8** | 7.8 | **23.6** | **18.3** | **46.2** | **31.2** |
| WS-DGOD | Gaussian FasterRCNN + EMA + PL | 39.0 | 19.4 | 60.4 | 9.4 | 23.8 | 17.3 | 44.0 | 30.5 |
| WS-DGOD | Gaussian FasterRCNN + EMA + PL + Regul. | **41.7** | **22.3** | **62.1** | **11.2** | **25.3** | **18.9** | **45.9** | **32.5** |
| DGOD | Gaussian FasterRCNN | 36.2 | 18.2 | 61.3 | 7.3 | 18.4 | 15.9 | 41.9 | 28.4 |

## B.5 Qualitative Results

Figs. 17 and 18 show the qualitative comparison on dusk-rainy and night-rainy, respectively. Similar to the artistic image dataset, the baseline model had false negative detections, which were improved by EMA, PL, and regularization.

Table 15: Comparisons of AP50 at each class on night-rainy of the car-mounted camera dataset (Wu & Deng, 2022). The values of * and ** are from (Wu & Deng, 2022) and (Vidit et al., 2023), respectively.

| setting | method | bus | bike | car | motor | person | rider | truck | mAP |
|---------|--------|-----|------|-----|-------|--------|-------|-------|-----|
| Single-DGOD | FasterRCNN* | 22.6 | 11.5 | 27.7 | 0.4 | 10.0 | 10.5 | 19.0 | 14.5 |
| Single-DGOD | CDSD* (Wu & Deng, 2022) | 24.4 | 11.6 | 29.5 | **9.8** | 10.5 | **11.4** | 19.2 | 16.6 |
| Single-DGOD | CLIP-based augmentation** (Vidit et al., 2023) | 28.6 | **12.1** | 36.1 | 9.2 | **12.3** | 9.6 | 22.9 | 18.7 |
| Single-DGOD | Gaussian FasterRCNN | 20.4 | 7.7 | 31.0 | 0.5 | 6.8 | 5.6 | 21.3 | 13.3 |
| Single-DGOD | Gaussian FasterRCNN + EMA | **33.9** | 11.1 | **38.5** | 0.8 | 10.5 | 8.8 | **29.2** | **19.0** |
| SS-DGOD | Gaussian FasterRCNN + EMA + PL | 35.7 | 9.8 | **46.7** | 1.4 | 12.6 | 10.8 | **32.0** | 21.3 |
| SS-DGOD | Gaussian FasterRCNN + EMA + PL + Regul. | **37.0** | **10.3** | 46.3 | **2.8** | **12.9** | **12.0** | 31.8 | **21.9** |
| WS-DGOD | Gaussian FasterRCNN + EMA + PL | **38.6** | 11.3 | **47.9** | **2.9** | 13.4 | 11.2 | **32.1** | 22.5 |
| WS-DGOD | Gaussian FasterRCNN + EMA + PL + Regul. | 38.3 | **13.4** | 46.2 | 2.7 | **15.1** | **14.0** | 32.0 | **23.1** |
| DGOD | Gaussian FasterRCNN | 38.9 | 7.6 | 46.7 | 1.8 | 9.8 | 11.3 | 32.1 | 21.2 |

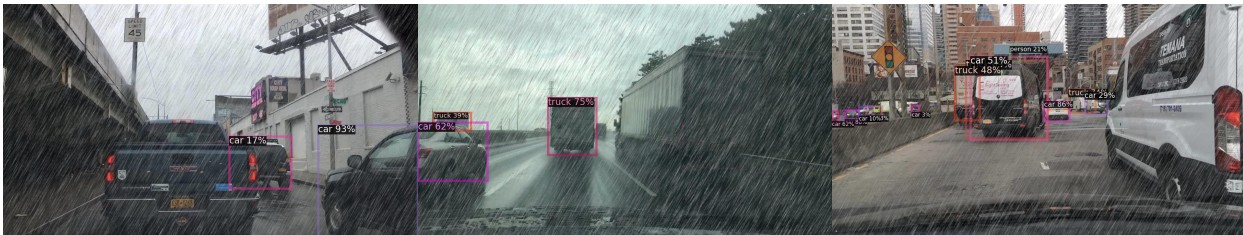

(a) Gaussian FasterRCNN trained on Single-DGOD setting (i.e., labeled data on daytime-sunny).

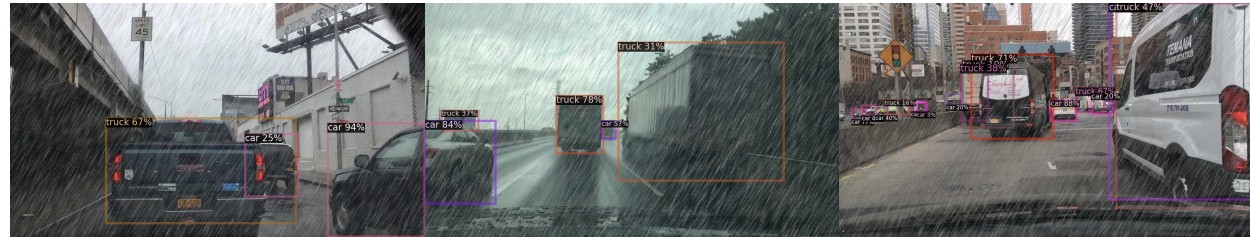

(b) Gaussian FasterRCNN + EMA + PL + Regul. trained on SS-DGOD setting (i.e., labeled data on daytime-sunny and unlabeled data on night-sunny and daytime-foggy).

Figure 17: Qualitative comparisons on dusk-rainy.

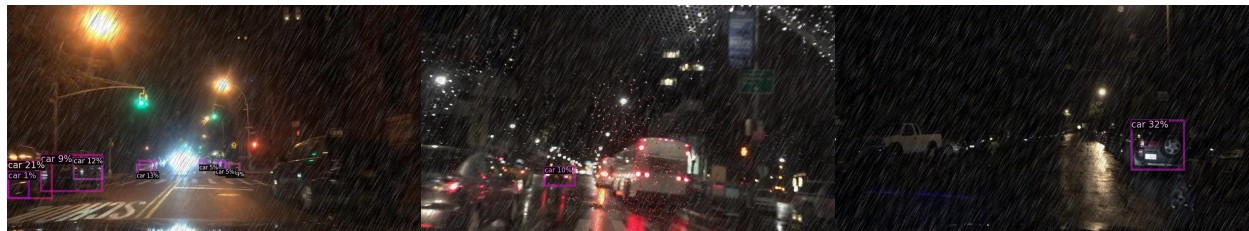

(a) Gaussian FasterRCNN trained on Single-DGOD setting (i.e., labeled data on daytime-sunny).

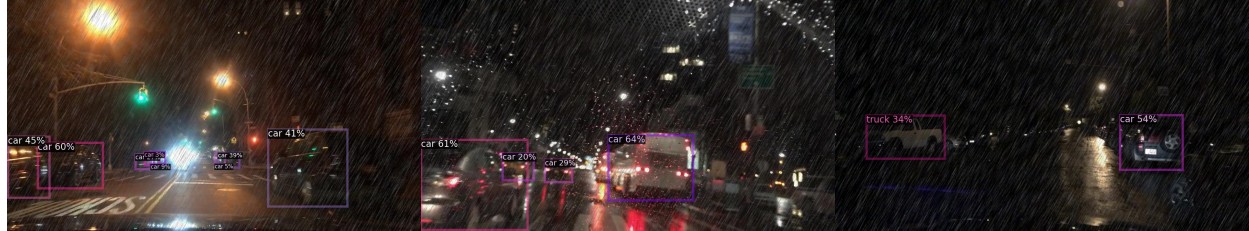

(b) Gaussian FasterRCNN + EMA + PL + Regul. trained on SS-DGOD setting (i.e., labeled data on daytime-sunny and unlabeled data on night-sunny and daytime-foggy).

Figure 18: Qualitative comparisons on night-rainy.

## C    Training Details

On the artistic style image dataset, the detectors were trained with 10,000 and 20,000 iterations for the pretraining and the student-teacher learning of SS-DGOD (or WS-DGOD), respectively. During the training, we saved the models and evaluated the performance on the validation at every 2,000 iterations, and the best model was used for the evaluation. The whole training took about one day. For fair comparisons, the compared models on Single-DGOD and DGOD were trained with 30,000 iterations, and the best models at the validation of every 2,000 iterations were used for evaluation.

On the car-mounted camera dataset, we performed the same procedure for training, validation, and evaluation, but the numbers of iterations for the pretraining and the student-teacher learning were set to 20,000 and 40,000 respectively, and the validation was conducted at every 4,000 iterations. The whole training took about two days. For fair comparisons, the compared models on Single-DGOD and DGOD were trained with 60,000 iterations, and the best models at the validation of every 4,000 iterations were used for evaluation.

## D    More discussions

### D.1    The Other Semi-supervised Domain Generalization Setting

There are two types of settings on semi-supervised domain generalization. The first setting assumes that only a part of the samples in each domain are labeled, similar to the previous works listed in Sec. 3.3. The other one assumes that only a part of the source domains are labeled (Lin et al., 2024). In this paper, we followed the previous SS-DGOD work (i.e., CDDMSL (Malakouti & Kovashka, 2023)) and tackled the second setting. To confirm the effectiveness of the Mean Teacher framework and the regularization on the other setting is one of our future works.

### D.2    Broader Impacts

In this work, we tackled the task of semi-supervised and weakly-supervised domain generalization for object detection (SS-DGOD and WS-DGOD), which are more practical settings than previous works. Also, we showed the good performance of the Mean Teacher learning framework, its interpretations, and a simple regularization method to boost the performance. Therefore, we believe that this paper has a potential positive social impact to enable practitioners or researchers to train robust object detectors to unseen domains in a simpler way than previous approaches. In addition, because Mean Teacher has been used across various tasks, our novel interpretation of why Mean Teacher becomes robust to unknown domains is likely to have a broad impact across a wide range of tasks. We are unable to identify any pertinent information concerning potential negative impacts.

## E    Reproducibility Statement

We submit the source code that can reproduce the results in this paper as a supplemental zip file.

