# OpenReview forum: "Seeking Flat Minima with Mean Teacher on Semi- and Weakly-Supervised Domain Generalization for Object Detection"
_TMLR — Withdrawn by Authors_

### Review · Reviewer_aVKH · 2025-03-03

**Summary Of Contributions:**

This paper explores two settings in object detection, i.e., semi- and weakly-supervised domain generalizable detection, finding that using the same Mean-Teacher learning framework can tackle these two tasks. Then, the authors give some interpretations of why the framework works and propose a simple regularization to achieve flatter minima. The proposed regularization can further boost the performance on both tasks.

**Audience:**

No

**Broader Impact Concerns:**

No potential ethical concerns.

**Claims And Evidence:**

Yes

**Requested Changes:**

1. The experiments are very weak and lack sufficient comparisons over state-of-the-art methods. Besides, the method gives a worse performance than AT, which was published in 2022.
2. Using a similar teacher-student pipeline for different cross-domain object detection settings has been fully studied in previous works, which is not the contribution of this work.
3. The technical contributions are weak, giving very similar ideas to previous works. The authors need to highlight the difference.

**Strengths And Weaknesses:**

Strength
- The paper is easy to follow and well organized.
- The high-level motivation of using the same Mean-Teacher pipeline to tackle different settings is interesting.

Weakness
- The paper lacks sufficient technical contributions. It has been fully explored using the Teacher-Student pipeline in both domain generalization and adaptation, no matter using the weakly-supervised or semi-supervised settings [1-2]. Hence, this is not a new observation.

[1] Shin, M. (2020, August). Semi-supervised learning with a teacher-student network for generalized attribute prediction. In European Conference on Computer Vision (pp. 509-525). Cham: Springer International Publishing.
[2] Deng, L., Zhang, X., & Shang, Z. (2020). Weakly supervised cross-domain mixed dish detection with mean-teacher. IEEE Access, 8, 201236-201246.

- The effectiveness of the proposed regularization is not well justified. In table 2, adding Regul. also gives a worse performance than the compared method AT (published in 2022) in the UDA-OD setting. Besides, regularization is not new and has similar effects to existing works [3]. The authors need to clarify the novelty and differences with previous works.

[3] Munir, M. A., Khan, M. H., Sarfraz, M., & Ali, M. (2021). Ssal: Synergizing between self-training and adversarial learning for domain adaptive object detection. Advances in neural information processing systems, 34, 22770-22782.

- The experiments are only conducted on a single type of domain shift, i.e., the artistic imaging. Can this method work well on different types of domain shifts, such as cross-weather (Cityscapes to Foggy Cityscapes) and cross-camera (Cityscapes to Kitti) domain shifts?

- The paper assumes that the teacher network generates sufficiently accurate pseudo-labels, but this assumption is not rigorously validated. Experiments analyzing the quality of pseudo-labels and their impact on performance are necessary.

- The paper lacks sufficient exploration of failure cases, particularly why pseudo-labeling sometimes degrades performance in certain settings (e.g., the dusk-rainy domain in the car-mounted camera dataset).

- The compared methods are out-of-date. The authors should give a comprehensive comparison with state-of-the-art methods published in these two years.

---

### Review · Reviewer_GAqN · 2025-03-12

**Summary Of Contributions:**

The paper shows that the Mean Teacher framework effectively trains object detectors for SS-DGOD and WS-DGOD settings, revealing its robustness to unseen domains due to flat minima in parameter space. It introduces a simple regularization to enhance this flatness and is the first to address the WS-DGOD setting.

**Audience:**

Yes

**Claims And Evidence:**

Yes

**Requested Changes:**

1. Update the baseline method to include more recent work.

2. Emphasize the contributions and novelty of the paper more clearly in the text.

3. Provide a clearer and more rigorous proof and explanation of the theoretical aspects.

**Strengths And Weaknesses:**

## Strengths

1. The author discovers a connection between the mean teacher framework and flat minima.

2. The author claims this paper is the first to address the WS-DGOD setting.

## Weaknesses

1. The baseline method is outdated and does not include works in recent one year.

2. The mean teacher approach has already been proven effective in many fields, such as Domain Adaptation (e.g., MIC [1]). It has also been widely applied in more challenging areas like TTA [2][3]. In the domain generalization field, related work has existed since 2021[4]. Therefore, I am concerned about the novelty of this work.

[1] MIC: Masked Image Consistency for Context-Enhanced Domain Adaptation

[2] CoTTA: Continual Test-Time Adaptation

[3] Exploring Sparse Visual Prompt for Domain Adaptive Dense Prediction

[4] Adversarial Teacher-Student Representation Learning for Domain Generalization

---

### Review · Reviewer_vjLV · 2025-04-01

**Summary Of Contributions:**

This paper proposed a weakly supervised domain generalization protocol for object detection task. This paper shows that both semi-supervised and weakly supervised domain generalisation can be solved in by a mean teacher framework. Explanations are provided that mean teacher learning leads to flat minima in the parameter space. An additional regularisation method is introduced to further enhance the flat minima during optimisation.

**Audience:**

Yes

**Broader Impact Concerns:**

No ethical concerns.

**Claims And Evidence:**

No

**Requested Changes:**

Evaluations on additional object detection frameworks are necessary.

**Strengths And Weaknesses:**

Strength:
1. The paper presents a well-structured motivation for seeking flatter minima and provides a theoretical foundation linking flatness to generalization performance.

2. The paper conducts multiple experiments on various datasets, comparing the proposed method against existing baselines and demonstrating its effectiveness.

Weakness:

1. While the paper introduces a novel method for seeking flat minima, it does not adequately compare it with alternative flatness-promoting approaches such as Sharpness-Aware Minimization (SAM) or entropy-based regularization techniques.

2. The theoretical analysis lacks a rigorous proof showing how the proposed method ensures better generalization. The presented arguments rely on intuition rather than formal guarantees.

3. The overall approach is simply reusing the well-known mean teacher architecture for the new setting. The weak labels are used to filter out the pseudo labels only.  The framework also resembles the MT approach for object detection proposed in Chen et al 2022. Therefore, the technical novelty is limited.

4. It is not very clear why the regularisation term could bring significant improvement in detection results. A more in-depth analysis of the results is required.

5. The effectiveness is currently evaluated on Faster RCNN only. It is necessary to demonstrate the effectiveness on a wide range of object detection frameworks, e.g. DETR, etc.

---

### Note · Authors · 2025-04-14

I have read and agree with the venue's withdrawal policy on behalf of myself and my co-authors.